# Conditional Independence Tests for Constraint-Based Causal Discovery: A Survey

## Abstract

Conditional Independence (CI) tests are the statistical engine of constraint-based causal discovery: in algorithms such as PC (Peter-Clark) and FCI (Fast Causal Inference), skeleton pruning and key orientations follow directly from CI decisions. This survey reviews CI testing with emphasis on assumptions, robustness, and scalability in high-dimensional and mixed-type settings common in biomedical domains. The survey organizes widely used CI methods into six families: partial-correlation, contingency-table, regression, nearest-neighbor, kernel, and machine-learning–based. Special emphasis is provided on the robustness layers that address the limitations of these families. For each family, the survey examines when CI decisions reflect the data-generating distribution and when they fail. By this, we link test-level properties, including power decay with conditioning set size and asymmetric type I/II error consequences, to graph-level errors in skeleton recovery and v-structure orientation. The survey also compares adoption across major R and Python libraries and summarizes open challenges, including mixed-type CI testing without discretization, small-sample error control, and strategies for improving scalability of CI-testing.

## 1 Introduction

Understanding the underlying causes of observed phenomena is essential in critical biomedical domains, where the ultimate goal is not only to predict outcomes but to design effective interventions (Li et al., 2015). In these fields, decisions based solely on correlations can be misleading or even harmful: a treatment may appear effective only because of hidden confounding factors, or a gene may seem predictive of disease severity without being a true driver of the biological process (Hernán & Robins, 2020). Causal discovery methods offer a way to uncover the structure of cause–effect relationships directly from data, making it possible to identify mechanisms rather than mere associations (Spirtes et al., 2000). For example, they can help determine whether a treatment influences recovery or is only associated with it because of patient characteristics. This shift from correlation to causation is particularly important at a time when the interpretability and fairness of AI systems are increasingly demanded in socially sensitive applications (Kusner et al., 2017). In healthcare, causal discovery methods help reveal underlying causal structure from data, enabling a better understanding of clinically meaningful relationships and supporting a range of healthcare applications (Scutari et al., 2017; Agrahari et al., 2018; Ganopoulou et al., 2021; Piccininni et al., 2020; Raghu et al., 2019; Ganopoulou et al., 2024; 2025; Krethong et al., 2008; Tangkawanich et al., 2008). This structural understanding can also provide a foundation for more informed, transparent, and interpretable decision-making. In biological research, the ability to uncover causal structure goes beyond mapping correlations among measured variables. Causal discovery enables scientists to infer which genes or proteins directly regulate others and to understand how these relationships change across cell types, developmental stages, or environmental conditions (Foroushani et al., 2017; Sachs et al., 2005; Boutsika et al., 2023; Skodra et al., 2023). These capabilities make causal discovery a promising technology for turning large, complex datasets into actionable insights.

Constraint-based causal discovery algorithms infer structure by systematically testing for conditional independencies among variables. For example, if two variables become independent after conditioning on a third variable, a constraint-based algorithm can use this as evidence that their apparent association is indirect rather than causal. At their core, these methods start from a fully connected graph and iteratively remove

edges when a conditional independence is detected between two variables given an appropriate conditioning set (Spirtes et al., 2000). Each decision to keep or remove an edge is driven by the outcome of a conditional independence (CI) test rather than by a global score or likelihood. Because of this, CI tests function as the core statistical mechanism of constraint-based approaches. Different tests encode different assumptions about the data (e.g., linearity, Gaussian noise, mixed types), and the choice of test influences the skeleton of the learned graph. In high-stakes biomedical settings, where incorrect edge removal or orientation can produce misleading causal claims and potentially harmful downstream decisions, selecting an appropriate CI test is therefore critical for reliable causal discovery. Beyond structure learning, CI tests also play an important diagnostic role in causal inference by allowing researchers to assess whether the independence relations implied by a manually specified Directed Acyclic Graph (DAG) are supported by the data (Shipley, 2000).

A distinctive advantage of constraint-based algorithms is their interpretability. Because they operate by explicitly testing conditional independencies and removing or keeping edges based on these tests, every step of the learning process is transparent and can be traced back to a concrete statistical decision. For practitioners in critical biomedical fields, this transparency makes it easier to audit why an edge was retained or removed and to build trust in the resulting causal graph.

Despite substantial progress in causal discovery, an important gap remains in how the literature treats CI testing. Existing surveys have reviewed causal discovery algorithms, applications, datasets, and software ecosystems from broad methodological perspectives, but CI testing is usually presented as one component among many rather than as the main organizing principle. As a result, researchers and practitioners still lack a focused synthesis of CI test families, their assumptions, robustness properties, computational trade-offs, and software support across major R and Python libraries. They also lack a systematic account of how test-level behavior, such as calibration, sensitivity to mixed-type data, or loss of power as conditioning sets grow, can propagate to graph-level errors in skeleton recovery and edge orientation.

**Contribution.** This survey addresses that gap by focusing on CI tests for constraint-based causal discovery from independent and identically distributed observational data. Score-based methods, continuous-optimization approaches, and hybrid procedures are outside our scope. The survey reviews the main CI test families used in constraint-based causal discovery, compares their implementation across major R and Python libraries, and examines how their assumptions, robustness, and computational properties shape their practical use. The survey also relates test-level properties to graph-level errors and provides practitioner-oriented guidance for selecting CI tests under different data-type, sample-size, and computational regimes. The survey targets researchers entering the area, particularly PhD students, as well as practitioners who need to identify the appropriate CI testing strategies for the characteristics of their data.

The remainder of this survey is organized as follows. Section 2 situates the present study within the existing literature and outlines its objectives. Section 3 outlines the theoretical foundations of causal discovery. Section 4 reviews major algorithms of constraint-based causal discovery, while Section 5 presents the main families of CI tests and their characteristics. Section 6 discusses practical considerations in CI testing including handling mixed-type data, statistical power, and test selection. Section 7 examines computational complexity and high-dimensional settings; Section 8 links algorithmic choices to transparency and trust; Section 9 showcases use cases in biomedical fields; Section 10 highlights unresolved challenges; and Section 11 summarizes key insights and points to future research directions.

## 2  Objectives of the Present Study

Several broad surveys have shaped the current understanding of causal discovery across domains. Vowels et al. (2022) provide foundational theory and a systematic overview of methods for learning causal graphs from both observational and experimental data, covering classical constraint-based and score-based approaches alongside newer continuous-optimization methods. Nogueira et al. (2022) expand the perspective to both causal discovery and causal inference, presenting a practical toolkit that includes algorithms, datasets, evaluation criteria, and software resources across multiple data modalities. Assaad et al. (2022) concentrate on time-series settings and offer an empirical comparison of Granger-causality, constraint-based, noise-based, score-based, and hybrid techniques, emphasizing that no single methodological family performs best across all temporal regimes. Zanga et al. (2022) synthesize theoretical and applied aspects of causal discovery by

distinguishing observational, interventional, and cyclic formulations and reporting on benchmark datasets and software implementations.

These reviews demonstrate the maturity of the field. Although multiple algorithmic families are now established, their assumptions, scalability properties, and support for mixed data types remains markedly inconsistent. This motivates a more targeted examination, such as the present survey, which focuses on CI tests for constraint-based discovery, the statistical component that enables one of the most interpretable branches of causal structure learning.

Relative to existing surveys, this work makes four contributions:

1. First, it provides a CI test-centric synthesis tailored to constraint-based causal discovery, treating CI testing as the primary design choice that governs skeleton pruning and edge orientation (Sections 4-6).

2. Second, it links test-level properties, i.e. assumptions, calibration, and power decay with conditioning set size, to graph-level failure modes, including spurious/missing edges and incorrect v-structure orientations (Sections 5, 6.2, and 7).

3. Third, it offers a comparative software view across major R and Python ecosystems, summarizing the availability of constraint-based algorithms and CI test families, as well as practical features such as conditioning set limits, logging, customization, parallelization, and prior-knowledge support. In doing so, it consolidates information that is otherwise scattered across documentation and vignettes (e.g., detailed guidance provided by MXM (Lagani et al., 2017)) into a single reference (Sections 4.1, 4.2, and 5; Tables 1–6).

4. Fourth, it translates these findings into practitioner-facing guidance via trade-off tables and heuristic decision frameworks for continuous, categorical, and mixed-type settings under different sample-size and computational regimes (Sections 6.3, 7, and 8; Table 5 and Figures 6–8).

To make these contributions self-contained, the survey briefly revisits where CI testing enters PC (Spirtes & Glymour, 1991), Grow-Shrink (Margaritis, 2003), and Fast Causal Inference (FCI) (Spirtes et al., 1995), and summarizes the orientation mechanisms needed to understand how CI test behavior propagates to edge-orientation errors and, ultimately, to graph-level reliability. This framing allows the survey to connect statistical properties of CI tests with the practical behavior of constraint-based algorithms in applied settings.

## 3 Foundations

### 3.1 DAGs and Bayesian Networks

A DAG is a graph $G = (V, E)$ consisting of a set of nodes $V$ and directed edges $E$ with the restriction that no directed cycles are present (Pearl, 1995). Each edge $X \to Y$ represents a direct relationship between variables $X$ and $Y$. The acyclicity condition guarantees that, when following directed edges from any node, one can never return to that node.

A Bayesian Network (BN) is a DAG in which the directed edges encode conditional dependencies. The joint distribution factorizes according to:

$$P(V) = \prod_{X \in V} P(X \mid \mathrm{Pa}(X)),$$

where $\mathrm{Pa}(X)$ are the parents of $X$, i.e., variables with directed edges pointing into $X$. The conditional independencies implied by a BN can be read directly from the graph via d-separation (Pearl, 1988). A path between two vertices is active given a conditioning set $Z$ if every node with converging arrows on the path is in $Z$ or has a descendant in $Z$, and every other node on the path is outside $Z$; otherwise the path is blocked. Two disjoint sets $X$ and $Y$ are d-separated given $Z$ when no active path connects a node in $X$ to a node in $Y$, in which case $X$ and $Y$ are conditionally independent given $Z$. In simple terms, d-separation identifies

when conditioning blocks all paths of influence between two variables, thereby rendering them independent. In this survey, BNs serve as the formal structure through which conditional independencies are expressed.

## 3.2 CPDAG and Markov Equivalence

Different DAGs can encode exactly the same set of conditional independence relations and are therefore indistinguishable based on purely observational data. Such DAGs are said to belong to the same Markov equivalence class. A Completed Partially Directed Acyclic Graph (CPDAG) (Andersson et al., 1997) represents this class by showing edges that are common to all equivalent DAGs as directed, and edges whose orientation cannot be determined from conditional independence information alone as undirected. The skeleton of a DAG is the undirected graph obtained by ignoring all edge orientations. All DAGs in a Markov equivalence class share the same skeleton and the same set of v-structures.

This phenomenon can be illustrated using three-node structures. Consider the chain and fork configurations:

$$X \to Z \to Y, \qquad X \leftarrow Z \to Y, \qquad X \leftarrow Z \leftarrow Y.$$

Although these three graphs differ in edge orientation, they encode exactly the same conditional independence relation:

$$X \perp Y \mid Z.$$

No other conditional independence holds among $X, Y, Z$; in particular, $X$ and $Y$ are dependent unless $Z$ is in the conditioning set. Because no set of CI tests can distinguish among these structures, they belong to the same Markov equivalence class and are represented in a CPDAG as an undirected chain

$$X - Z - Y.$$

In contrast, the v-structure $X \to Z \leftarrow Y$ encodes the single conditional independence

$$X \perp Y,$$

which holds marginally but fails once we condition on the common child $Z$. This pattern cannot be produced by any chain or fork configuration. As a result, the v-structure orientation is uniquely identifiable from observational data and appears as a directed structure in the CPDAG, even when other edges remain undirected.

## 3.3 Markov Blanket

The Markov blanket of a node in a Bayesian network is the minimal set of variables that renders that node conditionally independent of all remaining variables in the graph. Formally, the Markov blanket of a variable consists of its parents, its children, and the other parents of its children (often called "spouses"). The Markov blanket represents the complete local dependency structure of a variable, and plays a central role in several constraint-based causal discovery algorithms. These concepts form the theoretical basis for constraint-based algorithms, which infer causal structure by testing the conditional independencies implied by d-separation. Figure 1 illustrates the Markov blanket of a node $X$: once the values of its parents, children, and spouses are known, $X$ is conditionally independent of all remaining variables in the graph.

## 3.4 Assumptions in Causal Discovery

Causal discovery refers to the process of inferring directed relationships among variables from data under assumptions that permit causal interpretation (Spirtes et al., 2000). For constraint-based causal discovery, these assumptions include: (i) the Causal Markov condition, which ensures that each variable is independent of its non-descendants given its parents, equivalently, that every d-separation in the graph entails a conditional independence in the distribution; (ii) the Faithfulness condition, which ensures that the only conditional independencies in the distribution are those entailed by d-separation in the graph (the converse direction); (iii) Causal Sufficiency, which assumes the absence of unmeasured confounders; and (iv) correct

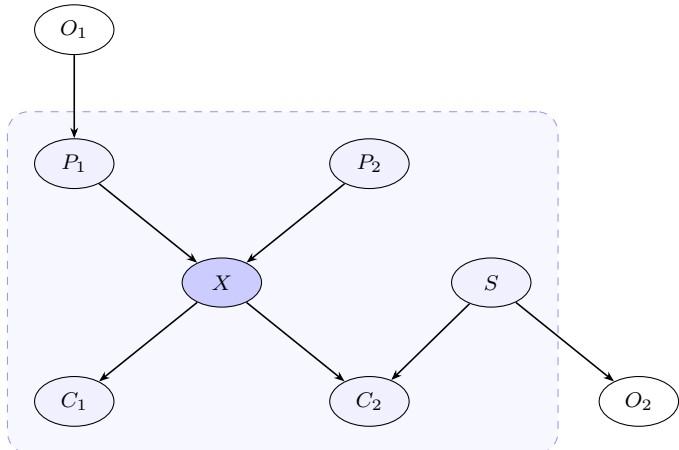

Figure 1: Markov blanket of node $X$ (darker shade, dashed rectangle): parents ($P_1$, $P_2$), children ($C_1$, $C_2$), and spouse ($S$, another parent of $X$'s child $C_2$). Nodes $O_1$ and $O_2$ lie outside the blanket.

statistical decisions, meaning that the inferred independence and dependence relations reflect those in the true underlying distribution. Under these conditions, observed statistical relations can be mapped to causal structure.

The core assumptions are outlined below, together with DAG-based illustrations demonstrating their implications.

### 3.4.1 Causal Markov Condition

Under the Causal Markov Condition, each variable is independent of its non-descendants given its parents.

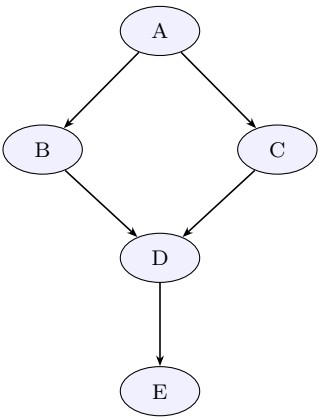

Figure 2: DAG illustrating the Causal Markov Condition.

In the DAG shown in Figure 2, this yields, for example, $B \perp C|A$, since $B$ and $C$ share the parent $A$, and are d-separated given $A$. It also implies $D \perp A|B, C$, because once the parents of $D$ are known, the upstream variable $A$ provides no further information about $D$. Similarly, $E$ is independent of $\{A, B, C\}$ given its parent $D$.

### 3.4.2 Faithfulness

Faithfulness requires that every conditional independence present in the distribution is entailed by a d-separation in the underlying DAG.

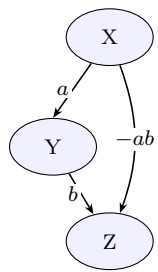

Figure 3: DAG illustrating a Faithfulness violation via path cancellation.

In the graph shown in Figure 3, there are two directed paths from $X$ to $Z$ ($X \to Y \to Z$ and $X \to Z$), so the DAG illustrates that $X$ and $Z$ should be dependent. Assume, as is standard for structural equation models, that the noise terms $\epsilon_Y$ and $\epsilon_Z$ are jointly independent and independent of the observed variables. Under the structural equations $Y = \alpha X + \epsilon_Y$ and $Z = \beta Y - \alpha\beta X + \epsilon_Z$, the contribution of $X$ to $Z$ cancels exactly, yielding $Z = \beta\epsilon_Y + \epsilon_Z$; since $\epsilon_Y$ and $\epsilon_Z$ are independent of $X$ by assumption, $Z$ is a function of noise alone and hence $X \perp Z$. This independence is not implied by any d-separation, and therefore the resulting distribution violates the Faithfulness assumption.

### 3.4.3 Causal Sufficiency

Causal sufficiency assumes that all common causes of the measured variables are themselves observed. Figures 4 and 5 illustrate two scenarios involving latent confounding and their impact on observed dependencies.

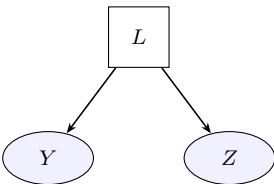

Figure 4: True causal structure with latent confounder $L$ influencing both $Y$ and $Z$.

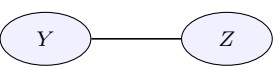

Figure 5: CPDAG inferred when $L$ is unmeasured: under (incorrectly assumed) causal sufficiency the dependence between $Y$ and $Z$ yields an unoriented edge $Y - Z$.

x

Figure 4 shows the true causal structure: the latent variable $L$ (represented as a square) is an unobserved common cause of $Y$ and $Z$. Figure 5 shows the observed structure that an algorithm assuming causal sufficiency might infer: because $L$ is unmeasured, the dependence between $Y$ and $Z$ is attributed to a undirected edge $Y - Z$, even though the true mechanism is confounding through $L$. Algorithms that allow latent confounding (e.g., FCI) can instead represent this ambiguity in a Partial Ancestral Graph (PAG) (Richardson, 1996; Zhang, 2008). A PAG represents an entire Markov equivalence class of causal structures that may contain latent confounders, using three kinds of edge endpoint mark (arrowhead, tail, and circle) to record the ancestral relations shared across the class. A circle denotes an endpoint whose orientation is not determined by the observed conditional independencies. This representation avoids the spurious causal claim.

### 3.4.4 Correct Statistical Decisions

Correct statistical decisions require that CI tests reliably detect the independence and dependence relations implied by the data-generating distribution. Formally, this assumption posits that the finite-sample CI test agrees with the population-level oracle - an idealization that no real test achieves. In practice, Type I errors (rejecting independence when it holds) introduce spurious edges into the skeleton, while Type II errors (failing to reject independence when it does not hold) remove true edges and may eliminate the separating sets needed for correct v-structure orientation, with cascading effects on subsequent Meek-rule applications

(Meek, 1995) (see Section 6.2 for a detailed analysis). This requirement holds both for algorithms assuming sufficiency (e.g., PC) and those allowing latent variables (e.g., FCI).

Among the four assumptions, correct statistical decisions is arguably the most practically fragile: the Causal Markov and Faithfulness conditions are properties of the data-generating process, whereas CI test accuracy is directly under the analyst's control through test selection, significance level, and sample size. This makes the choice of CI test the primary lever by which practitioners influence the reliability of constraint-based discovery.

## 4 Constraint-Based Causal Discovery Algorithms

Constraint-based algorithms form one of the oldest and most widely used families of causal discovery methods. They operate under the Causal Markov, Faithfulness, and Causal Sufficiency assumptions, together with the assumption of correct statistical decisions from CI tests (although algorithms that relax causal sufficiency also exist) (Spirtes et al., 2000). These methods start from a fully connected undirected graph and progressively remove edges when conditional independencies are detected between variables given suitable conditioning sets. Once a skeleton is obtained, orientation rules are applied to infer the direction of edges. Well-known examples include the PC algorithm, its latent-variable extension FCI, and their many variants such as PC-stable. Because every edge removal or orientation decision is tied to an explicit CI test, these algorithms provide a transparent and interpretable pipeline for learning causal structure. Their performance and reliability, however, are directly linked to the validity, power, and computational cost of the underlying CI tests, which makes understanding and comparing those tests central to advancing constraint-based causal discovery. Beyond the constraint-based setting, additive-noise-model methods such as RESIT (regression with subsequent independence test) (Peters et al., 2014) infer edge directions by testing the independence of regression residuals rather than of the observed variables.

### 4.1 Availability of Different Constraint-Based Algorithms in Python and R ecosystems

Widely used open-source packages for causal-graph structure learning include the R packages bnlearn (Scutari, 2010), pcalg (Kalisch et al., 2012), and MXM (Lagani et al., 2017), and the Python packages causal-learn (Zheng et al., 2024), pgmpy (Ankan & Textor, 2024), and gCastle (Zhang et al., 2021). In addition to general-purpose tabular causal discovery libraries, the Tigramite framework (Runge et al., 2019) provides constraint-based algorithms tailored for time-series data (e.g., PCMCI), together with a rich collection of CI tests. The availability of major constraint-based algorithms across these libraries is summarized in Table 1.

Table 1: Constraint-based algorithms across libraries (bnlearn 5.1, pcalg 2.7.12, MXM 1.5.5, gCastle 1.0.4, pgmpy 1.0.0, causal-learn 0.1.4.3; accessed 2026-02-15). ✓ indicates available implementation.

| Algorithm | bnlearn | pcalg | mxm | gcastle | pgmpy | causal-learn |
|---|---|---|---|---|---|---|
| PC-Stable (Colombo & Maathuis, 2014) | ✓ | ✓ | | ✓ | ✓ | ✓ |
| Grow-Shrink (Margaritis, 2003) | ✓ | | | | | |
| Incremental Association (Tsamardinos et al., 2003b) | ✓ | | | | | |
| Fast Incremental Association (Yaramakala & Margaritis, 2005) | ✓ | | | | | |
| Interleaved Incremental Association (Yaramakala & Margaritis, 2005) | ✓ | | | | | |
| Incremental Association with FDR (Peña, 2008) | ✓ | | | | | |
| Max–Min Parents and Children (Tsamardinos et al., 2003a) | ✓ | | ✓ | | | |
| HITON Parents and Children (Aliferis et al., 2010; 2003) | ✓ | | | | | |
| Hybrid Parents and Children (Gasse et al., 2014) | ✓ | | | | | |
| FCI (Spirtes et al., 1995) | | ✓ | | | | ✓ |
| PC (Spirtes et al., 2000) | | ✓ | ✓ | ✓ | ✓ | |
| RFCI (Colombo et al., 2012) | | ✓ | | | | |
| SES (Lagani et al., 2017) | | | ✓ | | | |
| PC (parallel) (Le et al., 2016) | ✓ | | | ✓ | ✓ | |
| CD-NOD (Huang et al., 2020) | | | | | | ✓ |

The following subsections describe the PC, FCI, and Grow–Shrink algorithms, highlighting where implementation choices differ across libraries.

## 4.2 The PC Algorithm

The PC algorithm, introduced by Spirtes & Glymour (1991), is a constraint-based causal discovery method that recovers a causal structure from observational data based on CI tests. The method assumes the Causal Markov condition, Faithfulness, Causal Sufficiency, and correct statistical decisions from CI tests.

The algorithm proceeds in two main phases. In the first phase, an undirected graph is estimated, starting from a fully connected graph and progressively removing edges when conditional independence is detected. For each pair of variables $(X, Y)$, CI tests of the form $X \perp Y \mid S$ are conducted, where $S$ is a conditioning set of increasing cardinality drawn from the adjacency set of $X$ and $Y$. If a set $S$ exists such that $X$ and $Y$ are independent given $S$, the edge $X - Y$ is removed from the skeleton and $S$ is stored in the separation set Sepset$(X, Y)$. This phase terminates when no further conditional independences can be identified or the maximum conditioning set size is reached.

In the second phase, the remaining undirected edges are oriented. First, all unshielded triples $X - Z - Y$, in which $X$ and $Y$ are non-adjacent, are examined. If $Z \notin$ Sepset$(X, Y)$, the triple is oriented as a v-structure: $X \rightarrow Z \leftarrow Y$. Subsequently, Meek's orientation rules (Meek, 1995) are applied. These are a set of four rules that iteratively orient as many of the remaining undirected edges as possible without creating a new v-structure or a directed cycle. Background knowledge can be incorporated by specifying forbidden edges or required directions.

The number of CI tests grows combinatorially with the maximum node degree $k$, leading to worst-case complexity exponential in $k$. Consequently, PC is computationally efficient for sparse graphs but becomes impractical when the true underlying causal structure is dense. Since orientation is polynomial, the dominant cost arises in the skeleton discovery step. The accuracy of PC strongly depends on CI test reliability; Type I and Type II statistical errors may propagate through v-structure orientation and subsequent Meek rule applications, potentially resulting in erroneous orientations.

The original PC algorithm is order-dependent, meaning that the order in which variables are processed can influence edge deletions and, consequently, the final graph. To address this, Colombo & Maathuis (2014) proposed the Stable-PC algorithm, which modifies skeleton discovery so that adjacency sets are updated only after all CI tests of a given conditioning level are completed. This ensures order-independence and improves reproducibility, particularly in finite samples.

Further improvements address computational scalability. Parallel-PC (Le et al., 2016) decomposes the CI testing workload across multiple processing units, significantly reducing runtime, especially in high-dimensional settings. Since most CI tests in the skeleton phase are independent, parallelization yields substantial efficiency gains without altering statistical decisions. Hybrid variants also exist, combining stability and parallel execution to achieve both reproducibility and scalability.

Implementation differences of the PC algorithm across major libraries, including support for stability, parallelization, and customization options, are summarized in Table 2.

## 4.3 FCI

The FCI algorithm, introduced by Spirtes et al. (1995), is a constraint-based causal discovery method designed to recover causal structure from observational data in the presence of latent confounders and selection bias. Like PC, FCI relies on CI tests, but it relaxes the assumption of Causal Sufficiency while retaining the Causal Markov condition, Faithfulness, and correct statistical decisions from CI tests.

FCI proceeds in multiple stages. In the first stage, the algorithm estimates an initial undirected graph, starting from a fully connected graph and iteratively removing edges based on CI tests of the form $X \perp Y \mid S$, where the conditioning set $S$ is drawn from the adjacency sets of $X$ and $Y$. As in PC, conditioning sets of increasing cardinality are considered, and whenever conditional independence is detected, the edge $X - Y$

Table 2: Support for key PC algorithm features across major causal discovery libraries, including stability variants, parallel execution, conditioning set limits, logging of CI test decisions, customization of CI tests, and incorporation of prior knowledge (package versions as in Table 1).

| Feature | causal-learn | gCastle | pgmpy | bnlearn | pcalg | mxm |
|---|---|---|---|---|---|---|
| Stable | ✓ | ✓ | ✓ | ✓ | ✓ | ✓ |
| Parallel | | ✓ | ✓ | ✓ | ✓ | ✓ |
| Max cond. set | | | ✓ | ✓ | ✓ | |
| CI test logs | ✓ | | | ✓ | ✓ | |
| Custom CI tests | ✓ | ✓ | ✓ | ✓ | ✓ | |
| Prior knowledge | ✓ | ✓ | ✓ | ✓ | ✓ | ✓ |

is removed and the corresponding separation set $\text{Sepset}(X, Y)$ is stored. This stage produces a preliminary skeleton, sometimes referred to as the FCI skeleton.

In contrast to PC, FCI does not assume that all common causes are observed. Therefore, additional CI tests are required beyond adjacency-based conditioning. In a subsequent refinement step, the algorithm considers Possible-D-SEP sets, which include nodes that may d-separate variable pairs through paths involving latent confounders. This step aims to remove spurious edges that remain due to unobserved variables and is critical for ensuring correctness under causal insufficiency.

Once the skeleton is finalized, FCI enters the orientation phase. As in PC, all unshielded triples $X - Z - Y$ are examined. If $Z \notin \text{Sepset}(X, Y)$, the triple is oriented as a v-structure $X \to Z \leftarrow Y$. However, due to the presence of latent confounders, not all edge orientations can be represented using directed edges alone. FCI therefore employs a richer edge mark vocabulary, allowing arrowheads, tails, and circles (e.g., $X \circ\!\!-\!\!\circ Y$, $X \to Y$, $X \leftrightarrow Y$) to encode causal ambiguity and potential confounding.

After initial v-structure orientation, an extended set of orientation rules, generalizing Meek's rules,is applied to propagate edge marks while preserving consistency with the observed CI relations and avoiding cycles or invalid causal interpretations. The final output of FCI is a PAG, which represents an equivalence class of causal graphs that are compatible with the data under latent confounding.

Background knowledge can be incorporated into FCI by specifying forbidden or required edge marks, allowing domain constraints to restrict admissible causal structures.

The computational complexity of FCI is significantly higher than that of PC. While skeleton discovery already exhibits combinatorial growth in the size of conditioning sets, the inclusion of Possible-D-SEP sets further increases the number and size of CI tests. In the worst case, complexity can grow super-exponentially in the number of variables, making FCI computationally demanding even for moderately sized graphs. As with PC, orientation steps are polynomial, and the dominant cost lies in CI testing.

The statistical reliability of FCI strongly depends on CI test accuracy. Errors in early CI decisions may lead to incorrect edge removals or erroneous v-structure orientations, with effects amplified by the more complex orientation rules and edge mark propagation. To mitigate these issues, several variants have been proposed. RFCI (Colombo et al., 2012) restricts the conditioning strategy to adjacency-based sets, trading some theoretical completeness for substantial gains in computational efficiency. Stable-FCI variants also exist (Colombo & Maathuis, 2014), adapting order-independent skeleton discovery principles to improve reproducibility in finite samples.

Implementation differences of FCI across major software libraries are summarized in Table 3, highlighting support for scalability options, CI test customization, and prior-knowledge integration.

### 4.4 Grow-Shrink

The Grow–Shrink (GS) algorithm (Margaritis, 2003) is a constraint-based causal discovery method designed to identify the Markov blanket of each variable using CI tests. Unlike global structure learning algorithms

| Feature | causal-learn | pcalg |
|---|:---:|:---:|
| Stable | ✓ | ✓ |
| Parallel | | ✓ |
| Maximum conditioning set | ✓ | ✓ |
| CI test result logs | ✓ | ✓ |
| Custom CI tests | ✓ | ✓ |
| Prior knowledge | ✓ | ✓ |

Table 3: Comparison of FCI implementation features in `causal-learn` and `pcalg`, including conditioning set controls, CI test customization, logging capabilities, and support for incorporating prior knowledge (package versions as in Table 1).

such as PC and FCI, GS focuses on local structure discovery by estimating, for each variable, the set of its parents, children, and spouses. The method assumes the Causal Markov condition, Faithfulness, Causal Sufficiency, and correct statistical decisions from CI tests.

The algorithm proceeds in two main phases: a grow phase and a shrink phase. For a target variable $X$, the goal is to identify its Markov blanket $\mathrm{MB}(X)$.

In the grow phase, the algorithm starts with an empty candidate set. Variables are iteratively added to the current Markov blanket candidate set if they exhibit marginal or conditional dependence with $X$ given the current set. Specifically, for each variable $Y \notin \mathrm{MB}(X)$, the algorithm performs a CI test of the hypothesis

$$X \perp Y \mid \mathrm{MB}(X)$$

and adds $Y$ to the candidate Markov blanket whenever the test rejects this independence (i.e., $X \not\perp Y \mid \mathrm{MB}(X)$). If dependence is detected, $Y$ is added to the candidate set.

The grow phase may introduce false positives, as some variables may appear dependent with $X$ only due to indirect associations. To address this, the shrink phase removes spurious variables. In this phase, each variable $Y \in \mathrm{MB}(X)$ is tested for conditional independence given the remaining candidates:

$$X \perp Y \mid \mathrm{MB}(X) \setminus \{Y\}$$

If conditional independence is detected, $Y$ is removed from the Markov blanket. This iterative removal continues until no further variables can be eliminated. The resulting set is taken as the estimated Markov blanket of $X$.

To learn a global graph, GS is applied independently to each variable, and local Markov blanket estimates are combined into an undirected skeleton. An edge between two variables $X$ and $Y$ is included if either $X \in \mathrm{MB}(Y)$ or $Y \in \mathrm{MB}(X)$. Edge orientation can then be performed using standard v-structure identification and orientation rules, similar to those employed in PC.

Grow-Shrink recovers each variable's Markov blanket with $O(p)$ conditional independence tests (the grow phase requires a constant number of passes over the $n$ variables and the shrink phase a single pass), so it learns the full structure with $O(p^2)$ tests overall. By contrast, the number of CI tests in PC can grow exponentially with the maximum node degree in the worst case. In sparse settings, GS scales well to high-dimensional problems, making it particularly attractive when the number of variables is large but the underlying graph is expected to be sparse. However, since GS conditions on potentially large candidate sets during the shrink phase, its performance may degrade when Markov blankets are large.

The accuracy of GS strongly depends on the reliability of CI tests. Errors in the grow phase may introduce false positives that propagate into the shrink phase, while false negatives in early tests may prevent relevant variables from ever being considered. Additionally, GS is order-dependent: the sequence in which variables are examined can influence both the composition of the Markov blanket and the resulting global structure.

Several extensions of GS have been proposed to improve robustness and scalability. Algorithms such as Incremental Association (IAMB) (Tsamardinos et al., 2003b), Fast-IAMB (Yaramakala & Margaritis, 2005), and Interleaved-IAMB (Yaramakala & Margaritis, 2005) refine the grow–shrink strategy by introducing statistical ordering and early stopping criteria. Max–Min Parents and Children (MMPC) (Tsamardinos et al., 2003a) and Hybrid Parents and Children (HPC) (Gasse et al., 2014) further enhance candidate selection by optimizing conditioning sets, leading to improved accuracy and reduced sensitivity to ordering effects.

Among the packages considered here, Grow–Shrink is primarily implemented in the bnlearn package.

## 5 CI Test Families

Throughout the paper, $n$ denotes the sample size, $p$ the number of observed variables, $|Z|$ the cardinality of the conditioning set in a given CI test, and $k$ the maximum vertex degree in the true graph.

CI testing lies at the core of constraint-based causal discovery algorithms such as PC, FCI, and their variants. Given random variables $X$, $Y$, and a (possibly multivariate) conditioning set $Z$, a CI test assesses the null hypothesis

$$H_0 : X \perp Y \mid Z$$

from finite observational data. The choice of CI test is critical: its assumptions, estimator properties, and calibration determine the correctness and stability of the learned causal structure.

This survey groups widely used CI procedures into six families. They are partial-correlation, contingency-table, regression, nearest-neighbor, kernel, and machine-learning–based. CI tests organized by (i) the dependence measure they target (e.g., correlation, likelihood, or conditional mutual information), (ii) the data types they support, and (iii) the statistical framework used for calibration. In addition to these families, several techniques act as robustness layers (e.g., shrinkage, permutation strategies, and test aggregation) that modify or wrap base tests rather than constituting distinct families. Powerful model-free measures of unconditional dependence, such as distance correlation (Székely et al., 2007) and Chatterjee's $\xi$ (Chatterjee, 2021), fall outside the scope of this survey, though conditional extensions of them can in principle serve as CI tests.

### 5.1 Partial-Correlation–Based Tests

Correlation-based CI tests form the classical foundation of constraint-based causal discovery for continuous data. Under multivariate normality, conditional independence between $X$ and $Y$ given $Z$ is equivalent to zero partial correlation.

The most widely used test in this family is Fisher's $Z$ test, which applies a variance-stabilizing transformation to the (partial) Pearson correlation coefficient and relies on the asymptotic normality of the transformed statistic. Let $\hat{\rho}_{XY \cdot Z}$ denote the sample partial correlation of $X$ and $Y$ given the conditioning set $Z$. Fisher's $Z$ test applies the variance-stabilizing transform

$$z(\hat{\rho}_{XY \cdot Z}) = \frac{1}{2} \ln \frac{1 + \hat{\rho}_{XY \cdot Z}}{1 - \hat{\rho}_{XY \cdot Z}} = \operatorname{arctanh}(\hat{\rho}_{XY \cdot Z}),$$

and, under the Gaussian null $\rho_{XY \cdot Z} = 0$, the rescaled statistic

$$\sqrt{n - |Z| - 3} \, \left| z(\hat{\rho}_{XY \cdot Z}) \right| \xrightarrow{d} \mathcal{N}(0, 1)$$

is compared against standard-normal quantiles. The statistical basis of the Fisher $Z$ transform was established by Hotelling (1953), while its extension to partial correlations and CI testing in Gaussian models is developed in classical multivariate analysis (Anderson, 2003). Under the null, the transformed statistic follows a standard normal distribution, yielding analytic $p$-values without resampling. With a precomputed sample covariance matrix, each test requires only $O(|Z|^3)$ operations for partial-correlation extraction via matrix inversion, making Fisher's $Z$ the fastest CI test in common use. It is implemented in all seven libraries

surveyed in Table 6, and serves as the default CI test in most constraint-based pipelines with continuous data.

However, the equivalence between zero partial correlation and conditional independence holds only under restrictive distributional assumptions. Baba et al. (2004) formalized the precise conditions under which partial correlation characterizes conditional independence, showing that violations of Gaussianity or linearity can invalidate correlation-based CI decisions even in the asymptotic regime.

Rank-based alternatives, such as Spearman's rank correlation (Spearman, 1987), can improve robustness under monotone nonlinear relationships, but zero partial rank correlation is not equivalent to conditional independence in general; hence these tests rely on additional structural assumptions and should be treated as approximate CI procedures (Baba et al., 2004).

## 5.2 Contingency-Table–Based Tests for Categorical Data

For categorical variables, CI testing is traditionally performed using contingency-table methods derived from multinomial likelihood theory and log-linear modeling of joint probability tables. These tests assess whether observed cell counts are consistent with the factorization implied by conditional independence.

The Pearson $\chi^2$ test and the likelihood-ratio test ($G^2$) are the most common representatives of this family. For categorical $X$ and $Y$ and a conditioning set $S$, let $n_{xyz}$ be the number of observations with $X = x$, $Y = y$, and $Z = z$, and let $n_{x \cdot z}$, $n_{\cdot yz}$, and $n_{\cdot \cdot z}$ denote the corresponding marginal totals within stratum $z$. The two statistics are

$$G^2 = 2 \sum_{x,y,z} n_{xyz} \log \frac{n_{xyz}\, n_{\cdot \cdot z}}{n_{x \cdot z}\, n_{\cdot yz}}, \qquad \chi^2 = \sum_{x,y,z} \frac{\left(n_{xyz} - \hat{m}_{xyz}\right)^2}{\hat{m}_{xyz}}, \qquad \hat{m}_{xyz} = \frac{n_{x \cdot z}\, n_{\cdot yz}}{n_{\cdot \cdot z}},$$

where $\hat{m}_{xyz}$ are the expected counts under conditional independence. Both are asymptotically $\chi^2$ with $\sum_s (r_x - 1)(r_y - 1)$ degrees of freedom, where $r_x$ and $r_y$ are the numbers of categories of $X$ and $Y$ (Agresti, 2002).

Finite-sample conditions and corrections for $\chi^2$ tests were studied by Cochran (1954), while Wilks' theorem establishes the asymptotic $\chi^2$ distribution of the $G^2$ statistic (Wilks, 1938). From an information-theoretic perspective, the $G^2$ statistic is proportional to conditional mutual information, linking likelihood-based and entropy-based CI testing (Cover, 1999). Both statistics are compared against a $\chi^2$ distribution with known degrees of freedom, providing analytic $p$-values. Per-test computation is $O(n)$, but the number of contingency-table cells grows exponentially with $|Z|$ and variable cardinality, causing sparse-table collapse well before computational limits are reached. At least one of the two tests is available in all seven surveyed libraries (Table 6).

Several variants have been proposed to address sparsity and small expected cell counts, including the Freeman–Tukey statistic (Freeman & Tukey, 1950), Neyman's modified $\chi^2$ test (Neyman, 1949), and the more general power-divergence family (Cressie & Read, 1984). A comprehensive treatment of these methods and their limitations under conditioning is provided by Agresti (2002). Despite these refinements, contingency-table CI tests suffer from rapid power degradation as the dimensionality of the conditioning set increases, due to table sparsity, unstable expected counts, and an explosion in degrees of freedom.

## 5.3 Regression-Based CI Tests

Regression-based CI tests assess conditional independence by fitting nested predictive models. In this framework, the null is typically operationalized as no additional predictive contribution of $X$ once $Z$ is included, most commonly via the hypothesis that the regression coefficient associated with $X$ is zero after adjusting for $Z$ under an assumed parametric model. This approach naturally accommodates a wide range of data types through the choice of a generalized linear model (GLM) (Agresti, 2002), which combines a response distribution, a linear predictor, and a link function relating that predictor to the mean response. Concretely, a GLM relates the conditional mean of $Y$ to $X$ and $Z$ through a link function $g$,

$$g\big(\mathbb{E}[Y \mid X, Z]\big) = \beta_0 + \beta_X X + \beta_Z^\top Z,$$

and the CI null is operationalized as $H_0 : \beta_X = 0$, i.e. $X$ carries no additional contribution once $Z$ is included. This is assessed with a likelihood-ratio statistic comparing the full model to the reduced model that omits $X$,

$$\Lambda = 2\big(\ell(\hat{\beta}_{\text{full}}) - \ell(\hat{\beta}_{\text{reduced}})\big) \xrightarrow{d} \chi^2_{\dim(X)},$$

or equivalently with a Wald or score test. Examples include linear regression for continuous outcomes (Kutner et al., 2005), logistic regression for binary outcomes (Hosmer et al., 2013), and ordinal regression models for ordered categorical variables (Williams, 2006).

Importantly, this regression-based notion is generally a statement about conditional mean independence rather than full conditional independence of distributions. Equality between a zero effect regression null and $X \perp Y \mid Z$ holds only under additional assumptions on the conditional distribution, such as correct specification of the link function and the functional form (how the predictors enter the linear predictor), absence of unmodeled interactions, and an error model in which removing the $X$ term eliminates all remaining dependence of $Y$ on $X$ given $Z$. When these assumptions fail, regression-based tests may still be useful as dependence screens, but their $p$-values need not be calibrated for CI and can induce systematic graph errors in constraint-based discovery. Conversely, when the GLM is correctly specified, including an appropriate link function and error distribution, testing whether the regression coefficient is zero is equivalent to testing full conditional independence within the assumed model.

Inference is typically performed using likelihood-ratio, Wald, or score tests, whose asymptotic validity is justified by classical likelihood theory (Wilks, 1938). However, regression-based CI tests rely critically on correct model specification. White (1982) showed that under misspecification, standard inferential procedures may converge to incorrect limits, leading to systematically biased CI decisions. In practice, violations arise from nonlinearity, heteroskedasticity, or distributional features not captured by the chosen model family.

To explicitly address mixed data types, Tsagris et al. (2018) proposed regression-based CI tests designed for constraint-based causal discovery with mixed variables. Likelihood-ratio testing has also been employed within mixed graphical models (Sedgewick et al., 2017). In practice, Tsagris et al.'s mixed-type regression CI tests are available in MXM (R), while Tigramite provides a regression-based CI test with automatic GLM family selection based on the response variable type. Nevertheless, fundamental limitations remain. Shah & Peters (2020) showed that conditional independence is not a testable hypothesis without further restrictions: any test whose type I error is controlled across all absolutely continuous distributions in the null has no power against any alternative. A useful CI test must therefore constrain the null through modeling assumptions, so this difficulty cannot be removed by regression modeling alone.

### 5.4 Nearest-Neighbor-Based CI Tests

Nearest-neighbor (KNN)–based CI tests estimate conditional independence by approximating conditional mutual information (CMI) using local density estimation derived from distances between observations. At the population level, conditional independence is equivalent to zero conditional mutual information,

$$X \perp Y \mid Z \quad \Longleftrightarrow \quad I(X; Y \mid Z) = 0.$$

KNN-based methods therefore operationalize CI testing by directly estimating this information-theoretic quantity in a nonparametric manner.

Runge (2018) introduced a KNN-based estimator of conditional mutual information that forms the basis for a flexible distribution-free CI test capable of detecting nonlinear dependencies. With $k$ nearest neighbors fixing a local length scale $\epsilon_i$ (in the maximum norm) around each sample $i$ in the joint space $\mathcal{X} \times \mathcal{Y} \times \mathcal{Z}$, the estimator is

$$\widehat{I}(X; Y \mid Z) = \psi(k) + \frac{1}{n} \sum_{i=1}^{n} \big[\psi(k_i^z) - \psi(k_i^{xz}) - \psi(k_i^{yz})\big],$$

where $\psi$ is the digamma function and $k_i^z$, $k_i^{xz}$, $k_i^{yz}$ count the points within distance $\epsilon_i$ of sample $i$ in the subspaces $\mathcal{Z}$, $\mathcal{X} \times \mathcal{Z}$, and $\mathcal{Y} \times \mathcal{Z}$, respectively. As the estimator has no tractable null distribution, significance is assessed with a (local) permutation test.

Subsequent work extended this approach to mixed continuous–categorical data and improved numerical stability for causal discovery applications (Popescu et al., 2025; Huegle et al., 2023). Specifically, CMIknn (Runge, 2018) is implemented in Tigramite, which also provides a mixed-type variant that extends the estimator to continuous–categorical data using an adapted distance metric (Popescu et al., 2025). Both versions use a permutation-based null distribution: the test statistic is recomputed $B$ times under random shuffles of $X$, yielding a nonparametric $p$-value at $O(Bn \log n)$ cost per test. mCMIkNN (Huegle et al., 2023) takes a different approach to mixed-type support by applying a discrete-aware rank transform that preserves ties for categorical variables, and is available as standalone Python code. Related estimators have also been proposed for mixed-type data, aiming to consistently approximate conditional mutual information without requiring discretization (Mesner & Shalizi, 2020; Zan et al., 2022).

Compared to parametric approaches, KNN-based CI tests make very weak distributional assumptions and can capture complex nonlinear relationships. However, they typically require larger sample sizes, are sensitive to the choice of neighborhood size and distance metric, and suffer from increased variance as the dimensionality of the conditioning set grows. Their computational cost is also substantially higher than that of correlation- or contingency-based tests, which can limit scalability in iterative constraint-based causal discovery algorithms.

## 5.5 Kernel-Based CI Tests

Kernel-based CI tests provide a flexible nonparametric approach to CI testing. Unlike classical parametric tests, they detect nonlinear conditional dependence without committing to a parametric functional form or Gaussianity, which makes them attractive for causal discovery in complex continuous-data settings.

A widely used representative is the Kernel Conditional Independence (KCI) test (Zhang et al., 2011). KCI evaluates whether two variables remain dependent after accounting for a conditioning set by comparing their residual dependence in a kernel-based feature space. Let $\widehat{\mathbf{K}}_{\ddot{X}|Z}$ and $\widehat{\mathbf{K}}_{\ddot{Y}|Z}$ be the centered kernel matrices of $X$ and $Y$ after regressing out $Z$ in the corresponding reproducing-kernel Hilbert spaces. The test rejects conditional independence for large values of

$$T_{\mathrm{KCI}} = \frac{1}{n} \operatorname{Tr}(\widetilde{\mathbf{K}}_{\ddot{X}|Z} \, \widetilde{\mathbf{K}}_{\ddot{Y}|Z}),$$

whose null distribution is a weighted sum of independent $\chi_1^2$ variables and is approximated in practice by a Gamma fit or a Monte-Carlo procedure. Its main advantage is broad flexibility, as it can detect a wide range of nonlinear dependencies under weak distributional assumptions. However, this flexibility comes at a substantial computational cost: KCI typically requires $O(n^3)$ time because of kernel matrix decompositions, which limits its practical use in larger samples. A Python implementation is available in causal-learn.

To address this scalability limitation, Strobl et al. (2019b) proposed two approximate kernel CI tests based on random Fourier features, namely the Randomized Conditional Independence Test (RCIT) and the Randomized Conditional Correlation Test (RCoT). These methods approximate kernel computations in a lower-dimensional random feature space, substantially reducing runtime while preserving much of the nonlinear modeling flexibility of kernel methods. RCIT and RCoT both have complexity $O(nd_f^2)$, where $d_f$ is the number of random features. Because $d_f$ is typically small, both methods scale approximately linearly in the sample size $n$, making them much more practical than KCI in large-sample settings. The original implementation is available in the R package RCIT.

In practice, kernel CI tests remain sensitive to tuning choices such as the kernel type, bandwidth selection, and, for randomized methods, the number of random features. Another important limitation is that standard implementations are mainly designed for continuous variables. Although mixed-type data can sometimes be accommodated through preprocessing or specialized kernel constructions, such support is not usually native or straightforward. Overall, kernel-based CI tests are powerful tools for detecting nonlinear conditional dependence, but their practical use involves a trade-off between flexibility, computational cost, and ease of application to mixed-type data.

### 5.6 Modern Machine-Learning–Based CI Tests

The Generalised Covariance Measure (GCM) of Shah & Peters (2020) tests conditional independence by regressing $X$ on $Z$ and $Y$ on $Z$ separately, computing the covariance of the resulting residuals, and testing whether it differs from zero. Writing $\hat{r}_i^X = X_i - \hat{\mathbb{E}}[X \mid Z_i]$ and $\hat{r}_i^Y = Y_i - \hat{\mathbb{E}}[Y \mid Z_i]$ for the residuals of the two regressions, the GCM statistic is

$$T_n = \frac{n^{-1/2} \sum_{i=1}^n \hat{r}_i^X \, \hat{r}_i^Y}{\left( n^{-1} \sum_{i=1}^n \left( \hat{r}_i^X \, \hat{r}_i^Y \right)^2 - \left( n^{-1} \sum_{i=1}^n \hat{r}_i^X \, \hat{r}_i^Y \right)^2 \right)^{1/2}} \xrightarrow{d} \mathcal{N}(0,1),$$

which is asymptotically standard normal regardless of the regression method used, provided the regressions are consistent for the conditional expectations. This model-agnostic property means that any supervised learning method (e.g., linear regression, random forest, gradient boosting) can serve as the regression backend, with the choice determining the test's effective assumptions: a linear backend recovers a test algebraically close to Fisher's $Z$, while a flexible backend can in principle detect nonlinear dependencies. Power depends directly on regression quality: poor residual estimation reduces the signal available for the covariance test. A weighted variant (WGCM) (Scheidegger et al., 2022) replaces the analytic normal calibration with a wild bootstrap, aiming to improve finite-sample behavior at the cost of additional computation. The GCM is implemented in the GeneralisedCovarianceMeasure R package; no surveyed Python library includes it natively.

More broadly, double machine learning (DML) constructs orthogonal estimating equations that yield valid inference even when nuisance functions are estimated using complex ML models (Chernozhukov et al., 2018). The key idea is cross-fitting: the sample is split into folds, nuisance parameters are estimated on held-out folds, and the orthogonal score is evaluated on the remaining data, avoiding the overfitting bias that would invalidate standard inference with flexible estimators. This framework has recently been extended to causal structure learning (Soleymani, 2024), where DML-based CI tests replace classical tests inside PC-style algorithms. While DML-based CI tests inherit the GCM's model-agnostic calibration guarantees, their empirical validation in iterative discovery pipelines remains limited.

Beyond regression-style approaches, recent work explores fully generative modeling strategies for CI testing. In particular, diffusion-model–based approaches assess conditional independence by modeling full conditional distributions using conditional score-based generative processes (Yang et al., 2025). While these methods offer substantial flexibility and can in principle capture arbitrary dependence structures, they require large sample sizes, careful training, and incur significant computational cost, making their integration into iterative discovery pipelines challenging.

### 5.7 Robustness Layers and Specialized Variants

Several CI tests have been developed to address specific violations of classical modeling or distributional assumptions. Rank-based permutation tests aim to mitigate the effects of discretization in mixed data (Dong et al., 2025), heteroskedastic CI tests relax constant-variance assumptions in regression-based approaches (Günther et al., 2022), ensemble CI frameworks aggregate multiple tests to improve robustness (Guan & Kuang, 2025), and discretization-aware CI tests explicitly model the impact of binning on independence decisions (Sun et al., 2025). While effective in targeted settings, these methods typically address only a single failure mode and do not resolve the fundamental challenges posed by high-dimensional conditioning.

Table 5 summarizes the main practical trade-offs across CI test families, focusing on assumptions, sample size demands, and scalability under large conditioning sets.

The availability of CI test families across major causal discovery libraries is summarized in Table 6.

## 6 Practical Considerations in CI Testing

Beyond understanding the theoretical foundations and algorithmic properties of different CI test families, practitioners face several concrete challenges when applying these tests in constraint-based causal discovery.

| Family | Distributional assumptions | Sample size needs | Large conditioning sets | High dimensionality | Runtime scaling |
|---|---|---|---|---|---|
| Partial correlation | Strong Gaussian | Small-medium n | Weak | Good | Very fast |
| Contingency-table | Discrete only | Moderate–large n | Weak | Poor | Fast-moderate |
| Regression-based | Weak–moderate | Moderate n | Moderate | Good | Moderate |
| Kernel-based | Very weak | Moderate-large n | Moderate | Moderate-good | Moderate-slow |
| Nearest-neighbor | Very weak | Large n | Weak | Weak | Slow |
| ML-based | Very weak | Large n | Moderate* | Moderate* | Slow |

Table 5: Comparison of CI test families across assumptions, sample size needs, and scalability. * ML-based tests are theoretically flexible but have limited empirical validation.

| CI Test Family | bnlearn | pcalg | mxm | gCastle | pgmpy | causal-learn | Tigramite[†] |
|---|---|---|---|---|---|---|---|
| Partial correlation | ✓ | ✓ | ✓ | ✓ | ✓ | ✓ | ✓ |
| Contingency-table | ✓ | ✓ | ✓ | ✓ | ✓ | ✓ | ✓ |
| Regression-based | | | ✓ | | | | ✓ |
| KNN / CMI | | | | | | | ✓ |
| Kernel-based | | | | | | ✓ | ✓ |
| ML-based | | | | | | | |

Table 6: Support of CI test families across causal discovery libraries (package versions as in Table 1; Tigramite 5.2.10.1). Regression-based tests are not implemented in `pcalg`, but are available via the `mxm` package and compatible with `pcalg` workflows. [†] Tigramite is primarily designed for time-series causal discovery but provides a rich collection of standalone CI tests (e.g., partial correlation, CMIknn, regression-based, and kernel-style tests) that can be used independently of temporal structure. No library surveyed provides a built-in ML-based CI test; DML-based and generative CI tests are available only as standalone research implementations, highlighting a gap between methodological development and practitioner tooling.

This section addresses three critical practical concerns: handling mixed-type data through discretization and alternative approaches (Section 6.1), understanding how statistical power and error control affect graph recovery (Section 6.2), and selecting appropriate CI tests based on data characteristics and computational constraints (Section 6.3).

## 6.1 Discretization in CI Testing

Discretization is one of the most commonly used strategies for enabling CI testing in the presence of mixed-type data. By mapping continuous variables to categorical representations, discretization allows the use of contingency-table–based CI tests, such as the $\chi^2$ and $G^2$ tests, which are simple, well studied, and computationally efficient (Wilks, 1938; Cover, 1999). As a result, discretization has been widely adopted in constraint-based causal discovery pipelines.

The appeal of discretization lies in its conceptual simplicity and broad applicability. Once variables are discretized, CI testing reduces to verifying the factorization of finite joint probability tables, yielding a clear and operational notion of conditional independence. However, discretization fundamentally alters the statistical object under consideration: the null hypothesis being tested no longer corresponds to conditional independence in the original continuous space, but rather to independence between discretized representations of the variables.

### 6.1.1 Discretization strategies

A wide range of discretization schemes has been proposed. Simple unsupervised approaches, such as equal-width or equal-frequency binning, are easy to implement but ignore dependence structure entirely. More principled methods aim to optimize information-theoretic or likelihood-based criteria. Notably, Hartemink's information-preserving discretization explicitly seeks discretizations that retain dependency and conditional independence relationships relevant for Bayesian network structure learning, rather than merely approximating marginal distributions (Hartemink & Gifford, 2001). Similarly, discretization based on the Minimum Description Length (MDL) principle integrates binning decisions directly into the structure learning objective, selecting discretizations that best trade off model fit and complexity (Friedman & Goldszmidt, 1996; Chen et al., 2017).

Despite these advances, most discretization methods still require choosing the number and placement of bins. As a result, discretization choices can vary substantially across studies, limiting reproducibility and complicating comparisons between causal discovery results.

### 6.1.2 Impact on CI testing and graph recovery

The effects of discretization extend beyond individual CI tests to the global properties of the learned causal graph. In constraint-based algorithms, CI decisions drive both edge removal and edge orientation through v-structure detection rules (Spirtes et al., 2000). Consequently, discretization-induced errors can propagate through the search procedure, leading to systematic distortions in the recovered graph structure. Empirical studies suggest that discretization can stabilize CI tests in very small samples by reducing variance, but often at the cost of attenuating weak, nonlinear dependencies that are critical for correct graph recovery. This effect has been observed in Bayesian network learning, where the choice of discretization method and number of bins significantly affects conditional probability tables and the inferred dependency structure, and no single method consistently preserves all aspects of the continuous relationships (Nojavan et al., 2017).

### 6.1.3 Discretization versus dependence-preserving alternatives

An important limitation of discretization is that, in general, it cannot preserve the full joint or conditional distribution of the data. This observation has motivated alternative approaches that aim to model dependence structure without explicit binning. Copula-based methods, for example, preserve rank-based dependence and enable CI testing for mixed data by modeling the copula separately from marginal distributions (Mesfioui & Quessy, 2008; Cui et al., 2016). Kernel-based CI tests offer another route, detecting dependencies

in high-dimensional feature spaces without discretization, albeit at significantly higher computational cost (Handhayani & Cussens, 2020).

## 6.2 Statistical Power and Error Control in CI Testing

Statistical power, the probability of correctly rejecting independence when it is false (i.e., detecting a true dependence), plays a central role in the reliability of CI testing within constraint-based causal discovery. Unlike classical hypothesis testing, where a single test is evaluated in isolation, CI tests are performed repeatedly as part of a sequential and adaptive graph search procedure. Errors therefore accumulate and propagate through the learning process, directly affecting skeleton recovery and edge orientation (Spirtes et al., 2000; Kalisch & Bühlmann, 2007).

### 6.2.1 Power decay with conditioning set size

A defining characteristic of CI testing in causal discovery is the rapid growth of conditioning sets. For most CI tests, statistical power deteriorates sharply as the cardinality of $Z$ increases, even when the underlying dependency structure remains unchanged. For partial correlation tests, the test statistic depends on $n-|Z|-3$ degrees of freedom; as $|Z|$ increases, variance increases and power drops quickly. (Anderson, 2003; Kalisch & Bühlmann, 2007). Analogous phenomena occur in contingency-table–based tests for categorical data, where increasing conditioning dimensionality leads to table sparsity, unstable expected counts, and substantial loss of power (Agresti, 2002).

### 6.2.2 Asymmetric impact of type I and type II errors

In constraint-based causal discovery, type I and type II errors have fundamentally different consequences. Rejecting conditional independence when it holds (type I errors) introduces spurious edges, whereas failing to reject it when it does not hold (type II errors) removes true edges and may prevent the identification of separating sets required for correct v-structure orientation. Because orientation rules depend critically on separating sets, type II errors can have cascading effects that eliminate entire classes of edge orientations, resulting in ambiguous or incorrect CPDAGs (Spirtes et al., 2000; Ramsey et al., 2006).

This asymmetry implies that classical significance levels alone are insufficient to characterize CI test reliability in causal discovery. Controlling the false discovery rate at the level of individual CI tests does not guarantee accurate recovery of the global causal structure (Colombo & Maathuis, 2014).

### 6.2.3 Finite-sample behavior and model misspecification

Finite-sample effects further complicate CI testing. Regression-based CI tests, such as those based on linear or GLM, rely on correct specification of functional form and error distributions; violations of these assumptions can lead to inflated type I or type II error rates and invalid asymptotic calibration (White, 1982; Tsagris et al., 2018). While rank-based and nonparametric CI tests are more robust to distributional assumptions, this robustness carries a cost in power: a CI test that stays valid across a broad nonparametric null cannot have power against every alternative, so a correctly specified parametric test is typically more powerful when its assumptions hold (Shah & Peters, 2020).

### 6.2.4 Expected power and graph-level performance

A key limitation of existing CI testing theory is the absence of tools for estimating expected power at the level of the causal graph rather than at the level of individual tests. Because CI decisions are interdependent and reused for edge orientation, the probability of correctly recovering a graph structure cannot be derived directly from per-test power alone. Existing theoretical guarantees are largely asymptotic and do not provide finite-sample, graph-level characterizations of performance (Ramsey et al., 2006; Li & Wang, 2009).

Recent work has shown that even mild heteroskedasticity can substantially reduce the power of classical CI tests and inflate false positives in PC-style algorithms, motivating variance-aware CI testing procedures (Günther et al., 2022).

### 6.2.5 Multiple testing correction

Constraint-based algorithms such as PC and FCI perform hundreds or thousands of CI tests during a single graph search, yet most implementations apply no explicit correction for multiple testing. The main reason is that $\alpha$ is set as a tuning parameter, not to control the family-wise error rate, the probability of one or more false edges across all the tests performed. In PC it is chosen to balance false against missing edges, and for consistency it is shrunk with the sample size, $\alpha_n \to 0$ as $n \to \infty$ (Kalisch & Bühlmann, 2007). Controlling the per-test type I rate at a conventional level does not, however, control the graph-level error rate: in finite samples, and especially for sparse graphs, the many CI decisions make spurious edges a genuine multiple-testing concern (Li & Wang, 2009), and asymptotic consistency alone does not characterise finite-sample error (Tsamardinos & Brown, 2008). Standard corrections are also ill-suited as applied directly to this test sequence: a Bonferroni correction divides the significance level by the number of tests (Dunn, 1961), so over PC's combinatorially large worst-case test count it becomes far too conservative; Benjamini–Hochberg, meanwhile, controls the FDR only for independent test statistics (Benjamini & Hochberg, 1995), and the CI tests in PC are neither independent nor known to be positively dependent, since hypotheses such as $X \perp Y_1 \mid Z$ and $X \perp Y_2 \mid Z$ share variables (Li & Wang, 2009). Where explicit control is desired, the natural target for this exploratory setting is the false discovery rate, controlled by the Benjamini–Yekutieli procedure (Benjamini & Yekutieli, 2001), as in FDR-controlled PC with edge-specific $p$-values (Strobl et al., 2019a), false-discovery bounds for local learning (Tsamardinos & Brown, 2008), and one-stage FDR control during skeleton discovery (Li & Wang, 2009); bnlearn also provides an IAMB-FDR variant (Peña, 2008). These variants provide explicit error-rate guarantees that the uncorrected default does not.

These considerations identify statistical power as a central bottleneck in constraint-based causal discovery. The interaction between conditioning dimensionality, finite samples, and CI test assumptions implies that improving CI test power, without inflating false positives, remains a key open challenge. Addressing this challenge is essential for developing causal discovery methods that are both statistically reliable and scalable to modern, high-dimensional applications.

### 6.3 Guidelines for CI Test Selection

The preceding sections examined individual CI test families (Section 5) and the role of power and error asymmetry (Section 6.2) in isolation. In practice, these factors interact: the choice of CI test must jointly account for data type, expected conditioning set depth, sample budget, and computational constraints. Rather than restating per-family properties, the survey highlights the major trade-offs that are most consequential for practitioners.

The primary axis of CI test selection is the match between test assumptions and data characteristics. When assumptions hold (e.g., Gaussianity for Fisher's $Z$, adequate cell counts for $G^2/\chi^2$), parametric tests offer the best power-to-cost ratio. When assumptions are uncertain, the practitioner faces a bias–variance trade-off: flexible tests (kernel, KNN-CMI) reduce model-misspecification bias but inflate variance at high $|Z|$, while regression-based tests offer intermediate flexibility at the cost of sensitivity to link-function choice (Tsagris et al., 2018).

A second practical axis is the interaction between conditioning set depth and sample size. Because power degrades across all families as $|Z|$ grows (Section 6.2.1), practitioners should consider capping $|Z|$ or using local discovery strategies (e.g., Markov blanket methods) when $n$ is small relative to the expected maximum conditioning depth. This decision is algorithmic rather than purely statistical and should be made jointly with the CI test choice.

Finally, transparent reporting of CI test choices, significance levels, maximum conditioning set sizes, and key separating sets is essential for reproducibility. The asymmetric impact of type II errors (Section 6.2.2) makes it particularly important to document edge-deletion decisions, enabling post-hoc auditing of potentially fragile CI conclusions.

These cross-cutting considerations underscore that CI test selection is not a purely technical detail but a central design choice that directly shapes the accuracy, interpretability, and scalability of constraint-based causal discovery.

To summarize these practical trade-offs, Figures 6, 7, and 8 present heuristic decision frameworks for selecting CI tests in continuous, categorical, and mixed data settings, respectively. These diagrams translate the theoretical considerations discussed above (distributional assumptions, conditioning set size, sample size, and computational budget) into practical guidance for method selection.

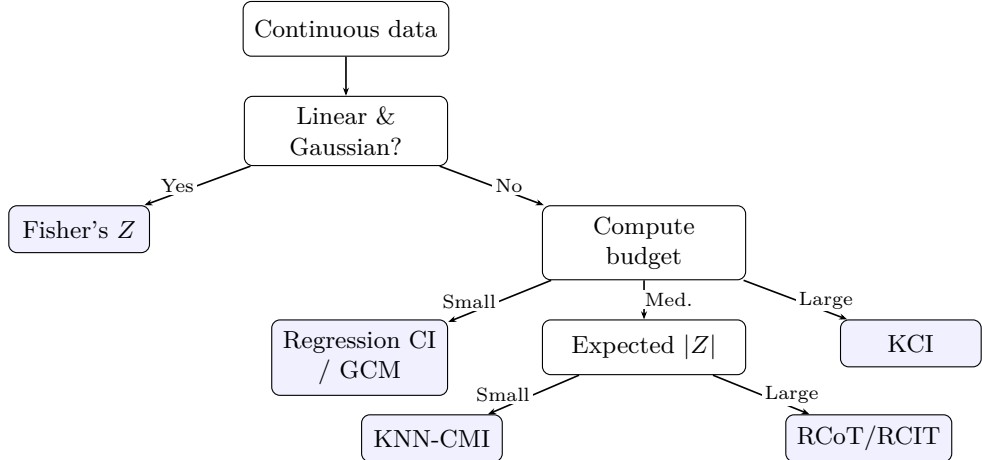

Figure 6: Heuristic decision diagram for selecting CI tests for continuous variables. The conditioning set split under medium budget reflects empirical evidence that KNN-CMI is better calibrated at small $|Z|$, while RCoT/RCIT achieve higher power at large $|Z|$ (Runge, 2018).

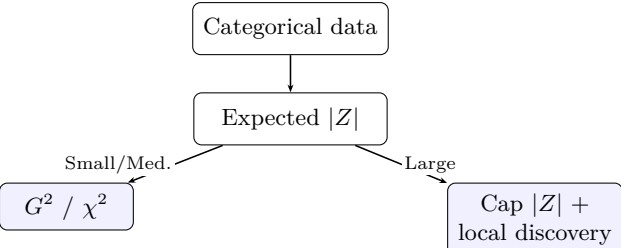

Figure 7: Heuristic CI test selection for categorical variables. For small-to-moderate conditioning sets, $G^2/\chi^2$ tests are effective provided expected cell counts remain adequate (Agresti, 2002). For large $|Z|$, table sparsity causes rapid power loss, motivating conditioning set limits or local discovery strategies such as Markov blanket methods.

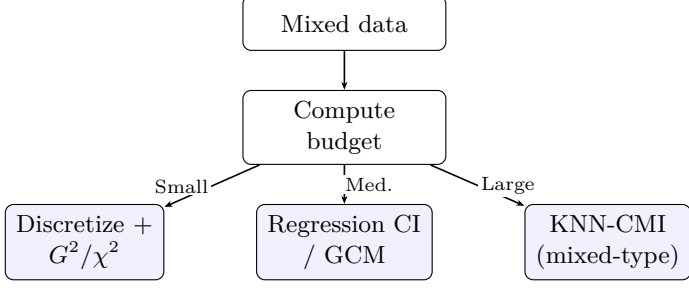

Figure 8: Heuristic CI test selection for mixed-type data. The small-budget discretization path is viable only for small-to-medium $|Z|$; when conditioning sets are large, regression-based CI tests (Tsagris et al., 2018) or GCM (Shah & Peters, 2020) are preferred even at higher cost. Mixed-type KNN-CMI variants (Popescu et al., 2025; Huegle et al., 2023) avoid discretization entirely but require larger samples and computational resources.

As an example, consider a practitioner with continuous, non-Gaussian data and a limited computational budget. Figure 6 directs them to regression-based CI tests or GCM as a practical compromise between the restrictive assumptions of Fisher's $Z$ and the computational cost of kernel methods. With a moderate budget, the diagram recommends approximate kernel tests (RCoT/RCIT) or KNN-CMI, depending on expected conditioning set depth: empirical evidence suggests that RCoT has slightly higher power at large $|Z|$, while KNN-CMI is better calibrated at small sample sizes (Runge, 2018). Only when computational resources are abundant does exact kernel CI testing (KCI) become practical, given its $O(n^3)$ cost (Strobl et al., 2019b). For categorical data (Figure 7), the primary consideration is conditioning set size: small-to-moderate sets permit standard $G^2/\chi^2$ tests, but large conditioning sets cause table sparsity and rapid power loss, motivating conditioning set limits or local discovery strategies. For mixed-type data (Figure 8), budget again governs the choice: discretization followed by contingency-table tests is the cheapest option but is limited to small-to-moderate $|Z|$; regression-based CI tests (Tsagris et al., 2018) and GCM (Shah & Peters, 2020) offer a middle ground; and mixed-type KNN-CMI variants (Popescu et al., 2025; Huegle et al., 2023) provide the most flexible nonparametric alternative at the highest computational cost.

These decision frameworks are based on theoretical properties and known limitations of each CI test family. Empirical validation across mixed-type data regimes, including systematic variation of sample size, conditioning set depth, and categorical proportion, is needed to confirm or refine these heuristics.

## 7 Scalability of CI-Based Causal Discovery

Constraint-based causal discovery (e.g., PC/FCI) can become computationally prohibitive as the number of variables grows. The bottleneck is the skeleton-learning phase, which performs many CI tests over conditioning sets of increasing size. In the worst case, the number of CI tests grows exponentially with the maximum neighborhood size (and thus can be exponential in $p$), although in sparse graphs the shrinking adjacency sets often reduce the practical workload substantially (Kalisch & Bühlmann, 2007).

### 7.1 Why high-dimensional settings are hard

When $p \gg n$, constraint-based causal discovery becomes difficult for both statistical and computational reasons. On the computational side, runtimes can grow rapidly because the algorithm must search over many candidate conditioning sets, and this combinatorial burden can dominate even when each CI test is relatively cheap. On the statistical side, limited sample size makes CI decisions less reliable. In particular, false independences (type II errors) are especially harmful because they may remove true edges early in the skeleton-learning phase and eliminate the separating sets needed for later v-structure orientations, thereby affecting the final CPDAG (Kalisch & Bühlmann, 2007).

A further practical complication is order dependence. During skeleton pruning, updates to adjacency sets can make the output of the original PC algorithm depend on the order in which variables or edges are processed. Stable-PC reduces this problem by freezing adjacency sets within each conditioning level and updating them only after all tests at that level have been completed. This improves reproducibility, but typically increases the number of CI tests and thus the runtime (Colombo & Maathuis, 2014).

To mitigate this additional cost, parallel-PC exploits the fact that, under stable-PC, CI tests within the same conditioning level can be carried out independently and distributed across CPU cores, yielding the same order-independent result while reducing computation time (Le et al., 2016). GPU-based implementations such as cuPC further accelerate this workload by offloading large batches of CI tests to CUDA-enabled hardware (Zarebavani et al., 2019).

### 7.2 Divide-and-conquer to reduce the number of tests

Beyond parallelising the same workload, newer methods aim to reduce the number of CI tests. A recent example is the Recursive Parallel Causal Discovery (RPCD) algorithm, which partitions variables, learns local skeletons, and then recursively merges components while re-applying PC-style pruning. This decomposition

can substantially cut the total CI tests compared to (stable) parallel-PC, yielding large runtime gains on high-dimensional benchmarks and real-world incident/gene-expression settings (Mondal et al., 2025).

Overall, scalability improvements fall into two complementary categories: (i) *systems-level acceleration* (multi-core parallelism, GPU batching) that keeps the algorithmic logic intact but speeds up CI testing; and (ii) *algorithmic restructuring* (divide-and-conquer, locality, pruning heuristics) that reduces the number of CI tests in the first place. The latter can deliver order-of-magnitude gains when meaningful problem decompositions exist, but may introduce additional design choices (e.g., partitioning strategy) that affect both runtime and graph accuracy (Mondal et al., 2025).

A complementary line of work addresses scalability by reducing the computational cost of individual CI tests rather than the total number of tests performed. The Ensemble Conditional Independence Test (E-CIT) (Guan & Kuang, 2025) is a divide-and-aggregate framework in which the data are partitioned into multiple subsets, a base CI test is applied independently to each subset, and the resulting $p$-values are aggregated using a combination rule.

Many widely used CI tests, particularly kernel- and distance-based methods, exhibit super-linear complexity in the sample size. By fixing the subset size, E-CIT reduces the effective per-test complexity to approximately linear in the total sample size. The framework is method-agnostic and can wrap existing CI tests as a plug-and-play layer, yielding substantial runtime improvements while preserving validity and, in many cases, improving empirical power in heavy-tailed settings.

## 8 Interpretability

In constraint-based causal discovery, each edge in the learned graph can be traced back to an explicit sequence of CI decisions, providing a transparent audit trail for why a relationship was retained, removed, or oriented (Spirtes et al., 2000; Kalisch & Bühlmann, 2007).

The interpretability of constraint-based methods, however, is inseparable from the properties of the underlying CI tests. CI tests with strong parametric assumptions may yield spurious independences when those assumptions are violated, leading to misleading edge deletions, while more flexible or non-parametric tests may uncover complex dependencies at the cost of increased computational burden. Consequently, the choice of CI test directly affects the stability and credibility of the resulting causal graph.

The degree of interpretability, however, varies across CI test families. Parametric tests such as Fisher's $Z$ or $G^2$ produce familiar statistics (partial correlations, likelihood ratios) whose magnitude and direction are directly meaningful to domain experts, making it straightforward to explain why a specific edge was removed. Regression-based CI tests offer similar transparency when the fitted model is a simple GLM: the coefficient, its standard error, and the $p$-value constitute a self-contained justification for each CI decision. By contrast, kernel-based and nearest-neighbor CI tests operate in high-dimensional feature spaces or rely on local density estimates that are difficult to summarize in human-readable terms. While these tests may detect subtler dependencies, the resulting edge decisions are harder to audit.

### 8.1 CI test choice and the audit trail

For the audit trail to be meaningful, the CI test must be interpretable to the intended audience. Reporting the test name, significance level, maximum conditioning set size, and key separating sets should be standard practice (Spirtes et al., 2000; Kalisch & Bühlmann, 2007). When nonparametric or ML-based CI tests are used, supplementary diagnostics, such as permutation $p$-value distributions or effect-size proxies, can help bridge the gap between statistical validity and human understanding.

### 8.2 Causal versus associational explanation

Predictive models for tabular data (e.g., gradient-boosted trees) excel at forecasting and feature ranking but remain fundamentally associational (Molnar, 2025). Post-hoc explainability methods such as SHAP attribute importance scores that reflect learned correlations rather than causal influence, and may assign high

importance to consequences or proxies rather than true causes. In contrast, each edge in a constraint-based causal graph is justified by a concrete CI decision, enabling interventional reasoning that feature-attribution scores alone cannot support (Pearl, 2009). This structural form of explanation targets *epistemic* questions: *Why does this outcome occur?*, rather than purely *predictive* ones: *Why did the model output this value?* In biomedical domains, this distinction is critical: epistemic explanations are required to anticipate the effects of interventions, which purely associational models cannot.

# 9 Applications of Causal Discovery in Biomedical Domains

Causal discovery and Bayesian network modeling have a long history of application in biomedical domains, where the primary objective is often to understand underlying mechanisms rather than to maximize predictive accuracy. In these domains, data are typically observational, high-dimensional, and of mixed type.

## 9.1 Gene expression and molecular network analysis

One of the earliest and most influential applications of causal discovery in bioinformatics is the analysis of gene expression data. Friedman et al. (2000) demonstrated how Bayesian networks can be used to model gene regulatory relationships in yeast using large-scale mRNA expression measurements, establishing a foundation for probabilistic causal modeling of molecular systems. Since then, causal and Bayesian network approaches have been applied extensively to transcriptomic datasets to distinguish direct from indirect dependencies and to uncover regulatory structure beyond simple correlation (Friedman, 2004; Markowetz & Spang, 2007).

Causal discovery has also been applied to benchmark and real-world biological datasets involving interventions and perturbations. The protein-signaling study of Sachs et al. (2005) remains a canonical example, where causal networks were inferred from single-cell data under multiple experimental conditions, demonstrating how interventional information can validate and refine causal graph structure. In gene expression settings, the choice of CI test is particularly consequential. Raw RNA-seq expression is count data (non-negative integers), and microarray measurements are continuous intensities; in practice counts are log- or variance-stabilising-transformed and then analysed as continuous (Law et al., 2014; Love et al., 2014), though often non-Gaussian and high-dimensional. This favours partial-correlation or nonparametric tests while making contingency-table approaches impractical without discretization, and it is a concrete instance of the mixed-type and discretization issues discussed in Section 6.1.

## 9.2 Hematological malignancies and disease-specific networks

In the context of cancer biology and hematological disorders, Bayesian and causal network models have been used to analyze gene expression profiles and identify disease-specific regulatory patterns. Agrahari et al. (2018) applied Bayesian network models to gene expression data from acute myeloid leukemia (AML) and myelodysplastic syndrome (MDS), illustrating how probabilistic causal models can support disease classification and subtype discrimination. Related large-scale network analyses have highlighted the role of extracellular matrix pathways and transcriptional regulators in AML, providing biologically interpretable insights into disease mechanisms (Foroushani et al., 2017). These studies typically rely on score-based or hybrid learning; when constraint-based methods are applied, the combination of mixed variable types (continuous expression values, categorical subtypes) and small patient cohorts places heavy demands on CI test robustness.

## 9.3 Neurodegenerative disease and systems-level analysis

Causal and network-based approaches have also been employed to study complex neurodegenerative diseases. Zhang et al. (2013) used an integrated systems biology framework to identify genetic nodes and networks associated with late-onset Alzheimer's disease, combining gene expression data with network analysis to reveal key molecular drivers. While such studies do not establish definitive causality, they demonstrate how causal discovery can guide hypothesis generation and prioritize candidate mechanisms for further experimental validation. In such high-dimensional, low-sample settings ($p \gg n$), CI test power decay with conditioning

set size is a primary concern, motivating the use of local discovery strategies and conservative conditioning set limits.

### 9.4 Healthcare data and clinical decision support

Beyond molecular biology, Bayesian networks have been applied directly to clinical datasets. Scutari et al. (2017) analyzed malocclusion data using Bayesian networks, illustrating how causal graphical models can capture complex dependencies among clinical variables and support interpretable decision-making in healthcare settings. In such applications, graphical models offer transparency that is difficult to achieve with black-box predictive methods, allowing clinicians to inspect and reason about inferred relationships. Clinical datasets frequently combine continuous lab values, binary indicators, and ordinal scores, making CI test selection for mixed-type data (Section 6) a recurring practical challenge.

### 9.5 Standard benchmark networks

Several benchmark Bayesian networks have become de facto standards for evaluating causal discovery algorithms. The ASIA network (Lauritzen, 1988), an 8-node model of lung disease diagnosis, is the simplest and most frequently used. The ALARM network (Beinlich et al., 1989) (37 nodes, 46 edges) models patient monitoring in an intensive care unit and is widely used for medium-scale evaluation. The INSURANCE network (Binder et al., 1997) (27 nodes, 52 edges) models car insurance risk assessment and is notable for its mix of discrete variable types. The Sachs protein-signaling network (Sachs et al., 2005) (11 nodes, 17 edges) is unique among standard benchmarks in that interventional data are available, enabling validation of edge orientation. At larger scale, the DREAM gene regulatory network challenges (Marbach et al., 2010) provide in silico and real expression datasets with known ground-truth networks, supporting evaluation of scalability and robustness. More recently, the CausalChambers project (Gamella, 2025) introduced physical laboratory devices with computer-controlled variables and known causal structure, providing a real-world testbed in which variables can be set by direct physical intervention. The Sachs network, the DREAM datasets, and the CausalChambers project thus complement the purely synthetic ASIA, ALARM, and INSURANCE networks by grounding evaluation in real or controlled data with known structure.

### 9.6 From association to intervention

Across these applications, a common theme is the transition from purely associational analysis toward models that support interventional and counterfactual reasoning. While causal discovery alone does not replace randomized trials, it provides a principled framework for integrating observational data, domain knowledge, and experimental evidence. In biomedical domains, causal graphs are therefore best viewed as tools for mechanistic insight and hypothesis generation, whose conclusions must be interpreted in light of underlying assumptions and validated through targeted experiments. Notably, few of the studies cited above report which CI test was used or justify its selection, a gap that underscores the need for the practical guidance developed in earlier sections of this survey.

## 10 Open Problems

Despite significant progress in CI testing and constraint-based causal discovery, several important challenges remain open.

### 10.1 Mixed-Type Data and Robust CI Testing

Most CI tests are tailored to either purely continuous or purely categorical variables. Real-world datasets often mix binary (e.g., mutation (present, absent)), ordinal (e.g., disease stage (A, B, C)), count (e.g., number of admissions), and continuous (e.g., hemoglobin) features. For example, large-scale hospital registries and cancer biobanks combine laboratory measurements with ICD codes and treatment flags. Developing unified, distribution-free CI tests that handle mixed data types without ad hoc discretization or transformation is a pressing research direction (Tsagris et al., 2018).

## 10.2  Small-Sample Settings and Controlling Error Rates

Many high-stakes applications provide only a few dozen samples but thousands of variables. In rare-disease cohorts or early-phase clinical trials, each patient is precious; in single-cell RNA-seq experiments, thousands of genes may be measured across only a few cell types of interest. Under these conditions, even theoretically sound CI tests can exhibit high variance and inflated false discovery rates. There is a need for new multiple-testing corrections, bootstrapping strategies, or Bayesian shrinkage priors to stabilise decisions in small samples.

## 10.3  Robustness to Violations of Key Assumptions

Constraint-based algorithms rely on assumptions such as the Causal Markov and Faithfulness conditions, Causal Sufficiency, and correct statistical decisions (Ling et al., 2025). When these assumptions are violated, skeleton discovery can become unreliable. Research into diagnostics and robustness measures that explicitly quantify assumption violations is still at an early stage.

## 10.4  Integration with Domain Knowledge and LLM-Assisted Orientation

Many algorithms output only a CPDAG or PAG with undirected or partially oriented edges. Expert input can often resolve these ambiguities, but manual annotation does not scale. Emerging approaches that combine domain priors with large language models (LLMs) show promise for proposing edge orientations by mining published knowledge and translating it into constraints for the learning process (Le et al., 2024; Ban et al., 2025). For such pipelines to be scientifically defensible, LLM-derived statements should be treated as soft priors rather than ground truth, accompanied by auditable provenance and evaluated for robustness to erroneous or hallucinated constraints.

## 10.5  Empirical Validation of CI Test Selection Guidelines

The heuristic decision frameworks proposed in this survey (Figures 6–8) are grounded in theoretical properties and individual empirical findings from the CI testing literature. However, a systematic empirical comparison of CI test families across data types, sample sizes, conditioning set depths, and graph densities is still lacking. Existing benchmarking studies, such as Raghu et al. (2018), have evaluated causal structure learning algorithms on mixed data but compared only a small number of CI tests (e.g., Multinomial LRT and Conditional Gaussian). A comprehensive study would need to cover the full range of CI test families reviewed here, evaluate not only per-test metrics (calibration, power) but also graph-level outcomes (SHD, F1, orientation accuracy) when different CI tests are plugged into the same constraint-based algorithm, and systematically vary conditioning set depth and graph density. Validating or refining CI test selection guidelines through such controlled experiments on benchmark networks and semi-synthetic data with known ground truth remains an important open direction.

Addressing these open problems would substantially improve the reliability, interpretability, and scalability of constraint-based causal discovery in real-world settings.

# 11  Conclusions

Causal discovery promises more than accurate prediction: it aims to uncover mechanisms that support trustworthy, actionable interventions in domains where mistakes carry high cost. This survey argued that, within constraint-based methods, the decisive component is the CI test. This survey reviewed CI tests across six methodological families, partial-correlation, contingency-table, regression, nearest-neighbor, kernel, and machine-learning–based, examining their assumptions, calibration, and computational trade-offs, and emphasizing a simple but often overlooked point: graph quality is only as good as the validity of its CI decisions.

Across biomedical applications, these trade-offs are visible. Classical tests such as $\chi^2$, $G^2$, Fisher's $Z$, and rank-based correlations are fast and transparent but can produce incorrect decisions under distribution shift,

nonlinearity, or small samples. More flexible approaches, including kernel-based and nearest-neighbor methods, target broader classes of dependencies (often via conditional mutual information) but typically require larger samples and incur higher computational and tuning costs. Modern machine-learning–based procedures can further relax modeling assumptions, yet their practicality in iterative discovery pipelines depends on careful calibration, sample splitting, and compute budgets. In $p \gg n$ regimes, the number and depth of CI tests can make exhaustive procedures impractical; parallelization, restriction to local neighborhoods, and stability analyses are often necessary to preserve both feasibility and credibility.

Two cross-cutting themes emerged. First, explainability benefits when each edge decision is traceable to a CI test with explicit assumptions; reporting the CI test, the maximum conditioning set size, and key separation set choices should therefore be standard practice. Second, robustness tools, such as resampling, aggregation, and variance-aware corrections, are best viewed as *wrappers* around base CI tests that can mitigate specific failure modes but do not remove the fundamental difficulty of high-dimensional conditioning.

Looking forward, there are three priorities: (i) reliable CI testing for mixed-type data without ad hoc discretization; (ii) scalable discovery pipelines that reduce CI-test burden through parallelism and test-reduction strategies; and (iii) integration of domain knowledge, including auditable human-in-the-loop and retrieval-assisted workflows for orienting partially directed graphs.

For practitioners, the takeaway is to align the CI test to data type and expected functional form; cap conditioning set size and parallelize where possible; and document assumptions alongside results. For researchers, the agenda is to close the gap between flexible, high-power tests and their runtime costs, develop principled mixed-type procedures, and formalize orientation assistance in a way that is reproducible and transparent.

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
