# OpenReview forum: "Conditional Independence Tests for Constraint-Based Causal Discovery: A Survey"
_TMLR — Decision pending for TMLR_

### Review · Reviewer_SyJS · 2026-04-25

**Summary Of Contributions:**

This paper, titled "Conditional Independence Tests for Constraint-Based Causal Discovery: A Survey," provides a comprehensive review of conditional independence (CI) testing as the foundational engine for causal discovery algorithms like PC and FCI.

Strengths:
1. A Unified Framework for CI Test Families: It categorizes existing methods into six distinct families: partial-correlation, contingency-table, regression, nearest-neighbor, kernel, and machine-learning-based.

2. Linking Statistical Properties to Graph Errors: The paper explicitly connects test-level issues—such as power decay as conditioning sets grow—to specific graph-level failure modes like spurious edges or incorrect collider orientations.

3. Comparative Software Review: It consolidates information on CI test availability and algorithmic features across major R (e.g., pcalg, bnlearn, MXM) and Python (e.g., causal-learn, pgmpy, gCastle) libraries.

4. Practitioner Guidance: The authors provide heuristic decision frameworks and trade-off tables to help researchers select the most appropriate test based on their data type, sample size, and computational constraints

Weaknesses:
1. It's mainly a summary paper with limited novelty.

**Audience:**

Yes

**Audience Explanation:**

PhD students, as well as practitioners who need to identify the appropriate CI testing strategies for the characteristics of their data would be interested.

**Claims And Evidence:**

Yes

**Claims Explanation:**

It's a summary paper with no specific claims.

**Requested Changes:**

None

---

> ### Author Response · Authors · 2026-06-09
> **Response to Reviewer SyJS**
>
> We thank the reviewer for the positive assessment and for clearly summarizing the survey's contributions: “A Unified Framework for CI Test Families”, ”Linking Statistical Properties to Graph Errors”, “Comparative Software Review”, and “Practitioner Guidance”. We are also glad the reviewer found the work relevant to PhD students and practitioners.

---

### Review · Reviewer_fALU · 2026-05-18

**Summary Of Contributions:**

The paper provides a thorough review of CI tests and their use in causal discovery algorithms. This could be a nice resource to help practitioners more rigorously apply causal discovery methods.

**Audience:**

Yes

**Audience Explanation:**

This is a helpful review paper that collects a lot of information (e.g., relevant to practitioners interested in applying causal discovery methods) in one coherent place.

**Claims And Evidence:**

No

**Claims Explanation:**

A few details and references are missing, but it should be easy to fix.

**Requested Changes:**

Critical:
1. $d$-separation is a relatively important idea in the paper, but it's never actually defined (and a clear reference to a definition in the literature isn't even provide).
2. No reference is given for CPDAGs ([here](https://projecteuclid.org/journals/annals-of-statistics/volume-25/issue-2/A-characterization-of-Markov-equivalence-classes-for-acyclic-digraphs/10.1214/aos/1031833662.full) is a standard one)
3. In Section 3.2, I don't think it makes sense to list some and not all of the dependence statements. The usual thing to do is just talk about the CI statements. Also, it mentions 'collider orientation', but v-structure is more accurate?
4. At the beginning of 3.3, faithfulness is described as guaranteeing that "all and only" the CI relations in the data are implied by the graph. Isn't this including the Markov condition? Faithfulness is rather that the only CI relations in the data are those implied by the graph. and in 3.3.2, "correspond to" is ambiguous---Faithfulness is just one direction of the correspondence (and Markov is the other).
5. The DAG in Figure 4 isn't a CPDAG, so it wouldn't actually be recovered by the methods considered in this review. Maybe picking a different example would be less confusing?
6. "skeleton" is used without really being defined
7. "PAG" is used without being defined and without a reference/citation
8. Section 3.3.4 mentions "finite sample CI oracle", but I suppose this should rather be "test"? Also, better to use consistent wording instead of switching between "rejections of independence" and "reject false null hypothesis". Meek-rule is also undefined and has no reference here; "e.g.m" instead of "e.g.,"
9. no reference for stable-FCI in section 4.3
1. Table 3 should contain FCI algorithms, not PC, right?
1. Section 4.4 shows a dependence relation and calls it a CI test
1. What does it mean that the "complexity of GS is typically lower than that of PC"? I don't understand "typically" here; or maybe "run time" is meant instead of "complexity"?
1. no references for algorithms at end of Section 4
1. In Section 5.3, "link and functional form" isn't really explained; "GLM" isn't defined; "uniformly powerful CI test" isn't defined
1. no reference for Bonferroni or FDR correction
1. Section 9.1 claims that gene expression data is continuous; but raw expression data is usually count data (nonnegative integers)
1. Section 10.1 mentions sex as a binary variable, but that's not really the modern scientific understanding of it (for example, see [this recent article](https://link.springer.com/article/10.1186/s13293-025-00803-7), which also discusses the statistical issues that arise from such an assumption)

Minor:
1. In the first paragraph in the intro, why are "correlations between molecules" in particular mentioned as biological research? Seems obvious that a lot of biological research is concerned with larger scale phenomena than molecular interactions.
2. Could be worth mentioning [RESIT](https://jmlr.org/papers/volume15/peters14a/peters14a.pdf) and other additive noise model approaches that also use CI tests.
3. Methods should be given citations when they're introduced, like PC, GS, and FCI at the end of Section 2.
1. Many of the references are formatted incorrectly (capitalization like "Llm" and "markov"; "et al." instead of full lists of names; Pena instead of Peña, etc.)
1. Maybe swapping the sections 3.4 and 3.3 makes sense?
1. In section 4, be more clear that it starts at a complete undirected graph
1. I don't really get the "interpretability" argument repeated a few times about constraint-based methods. Doesn't this apply to GES too? Decomposability and local consistency of the score ensure each edge addition or removal has a clear interpretation, analogous to a CI test.
1. PC was introduced in 1991 by [this paper](https://journals.sagepub.com/doi/abs/10.1177/089443939100900106?casa_token=ylc-73q20-0AAAAA:8bkqCYpBMJOKEZE2J5g5P-etgpo02llzWZUn8ADff4AGD70FA_E3RmR2clefeBYoCLun5I2o4Gw)
1. Section 5.5 includes the tautology "they can capture nonlinear relationships without assuming linearity"
1. Inconsistent use of periods at the end of (sub(sub))section titles, e.g.., 6.2.5 vs 6.2.6
1. In Section 8.2, what does "evaluate counterfactual scenarios, and assess external validity" mean?
1. use \citet for the Agrahari ref in Section 9.2
1. 3 different real data benchmarks are mentioned in Section 9.5, but only the last is claimed to "complement purely synthetic benchmarks"?
1. How about mentioning other powerful model-free tests, like distance correlation and Chatterjee's correlation?

---

> ### Author Response · Authors · 2026-06-09
> **Response to Reviewer fALU: Critical Changes (Part 1)**
>
> We thank the reviewer for the careful reading and constructive feedback, and for noting that the survey is a useful, coherent resource for practitioners. The requested changes were about missing definitions and citations together with a small number of correctness and precision points, all of which we have addressed. Below, each reviewer comment is quoted, followed by our response and the action taken to address the review suggestion. The references suggested by the reviewer are now appropriately integrated in the manuscript. Because of the changes, Section, Figure, and Table numbers below refer to the revised manuscript.
>
> $\textbf{Comment 1.}$ $d$-separation is a relatively important idea in the paper, but it's never actually defined (and a clear reference to a definition in the literature isn't even provide).
>
> $\textbf{Response:}$ Thank you for the feedback.
>
> $\textbf{Action:}$ We now provide a formal definition of $d$-separation at its first use in Section 3.1. Additionally, we cite Pearl (1988), as the original source.
>
> $\textbf{Comment 2.}$ No reference is given for CPDAGs (here is a standard one).
>
> $\textbf{Response:}$  We thank the reviewer for the pointer.
>
> $\textbf{Action:}$ We now cite Andersson et al. (1997) in the definition of the CPDAG / Markov equivalence class in Section 3.2.
>
> $\textbf{Comment 3.}$ In Section 3.2, I don't think it makes sense to list some and not all of the dependence statements. The usual thing to do is just talk about the CI statements. Also, it mentions 'collider orientation', but v-structure is more accurate?
>
> $\textbf{Response:}$ Both are valid points, and we have acted on each.
>
> $\textbf{Action:}$ We have rewritten the three-node example to state the characterizing conditional independence relation directly ($X \perp Y \vert Z$ for the chain/fork; $X \perp Y$ for the v-structure) rather than a partial mix of dependence and independence statements, and we now use "v-structure" consistently throughout the paper (including the Abstract and Section 2, where we list the contributions of the paper).
>
> $\textbf{Comment 4.}$ At the beginning of 3.3, faithfulness is described as guaranteeing that "all and only" the CI relations in the data are implied by the graph. Isn't this including the Markov condition? Faithfulness is rather that the only CI relations in the data are those implied by the graph. and in 3.3.2, "correspond to" is ambiguous---Faithfulness is just one direction of the correspondence (and Markov is the other).
>
> $\textbf{Response:}$ Agreed; this was a genuine imprecision and we have corrected it.
>
> $\textbf{Action:}$ Faithfulness is now described as the single direction it represents. We have edited the beginning of Section 3.4 accordingly.
> In the new version we have removed the ambiguous "correspond to" wording.
>
> $\textbf{Comment 5.}$ The DAG in Figure 4 isn't a CPDAG, so it wouldn't actually be recovered by the methods considered in this review. Maybe picking a different example would be less confusing?
>
> $\textbf{Response:}$ That’s a valid point. Thank you very much.
>
> $\textbf{Action:}$ We have changed the edge to the undirected $Y − Z$ and revised the caption so the figure now shows the spurious CPDAG that would be inferred (see updated Figure 4 in Section 3.4.3).
>
> $\textbf{Comment 6.}$  "skeleton" is used without really being defined
>
> $\textbf{Response:}$ Thank you for the suggestion.
>
> $\textbf{Action:}$ We now define the skeleton at its first substantive use in Section 3.2.
>
> $\textbf{Comment 7.}$ "PAG" is used without being defined and without a reference\citation
>
> $\textbf{Response:}$ Good point. Thank you very much.
>
> $\textbf{Action:}$ We now define the Partial Ancestral Graph at first use and we cite Richardson (1996), who introduced PAGs, and Zhang (2008) for the more contemporary perspectives (Section 3.4.3).
>
> $\textbf{Comment 8.}$ Section 3.3.4 mentions "finite sample CI oracle", but I suppose this should rather be "test"? Also, better to use consistent wording instead of switching between "rejections of independence" and "reject false null hypothesis". Meek-rule is also undefined and has no reference here; "e.g.m" instead of "e.g.,"
>
> $\textbf{Response:}$ The reviewer is right on all points.
>
> $\textbf{Actions:}$
> - "finite-sample CI oracle" is now "finite-sample CI test";
> - the Type I / Type II discussion now uses consistent "reject independence" wording in all places it appears (Sections 3.3.4 and 6.2);
> - the Meek rules are now defined (Section 4.2) and cited (Meek 1995);
> - we have corrected the "e.g.m" typo.

---

> > ### Author Response · Authors · 2026-06-09
> > **Response to Reviewer fALU: Critical Changes (Part 2)**
> >
> > (Continuing our point-by-point response.)
> >
> > $\textbf{Comment 9.}$ no reference for stable-FCI in section 4.3
> >
> > $\textbf{Response:}$ Thank you for this suggestion.
> >
> > $\textbf{Action:}$ We now cite Colombo & Maathuis (2014) for the stable-FCI / order-independent variants in Section 4.3.
> >
> > $\textbf{Comment 10.}$ Table 3 should contain FCI algorithms, not PC, right?
> >
> > $\textbf{Response:}$ Thanks for noting this.
> >
> > $\textbf{Action:}$ We have renamed those rows to "Stable" and "Parallel," so the table now reports the stability and parallel-execution support of the FCI implementations (Table 3).
> >
> > $\textbf{Comment 11.}$ Section 4.4 shows a dependence relation and calls it a CI test
> >
> > $\textbf{Response:}$ Again, we are very thankful for this valid point.
> >
> > $\textbf{Action:}$ In the Grow phase of Grow-Shrink we now state the test as the conditional independence hypothesis $X \perp Y \mid \operatorname{MB}(X)$, and $Y$ is added to the candidate Markov blanket when the test rejects this independence (Section 4.4).
> >
> > $\textbf{Comment 12.}$ What does it mean that the "complexity of GS is typically lower than that of PC"? I don't understand "typically" here; or maybe "run time" is meant instead of "complexity"?
> >
> > $\textbf{Response:}$ Thank you for prompting us to make this statement more precise.
> >
> > $\textbf{Action:}$ Grow-Shrink recovers each variable's Markov blanket with $O(p)$ conditional independence tests in the number of variables p, hence $O(p^2)$ for the full structure, whereas PC's number of CI tests can grow exponentially with the maximum node degree in the worst case. We cite Margaritis (2003) for the complexity analysis (Section 4.4).
> >
> > $\textbf{Comment 13.}$ no references for algorithms at end of Section 4
> >
> > $\textbf{Response:}$ Thank you, we have added the missing references.
> >
> > $\textbf{Action:}$ We have now added references for IAMB and its variants, MMPC, and HPC at the end of Section 4.
> >
> > $\textbf{Comment 14.}$ In Section 5.3, "link and functional form" isn't really explained; "GLM" isn't defined; "uniformly powerful CI test" isn't defined
> >
> > $\textbf{Response:}$ Thank you for this helpful point.
> >
> > $\textbf{Action:}$
> > - We now introduce the generalized linear model (GLM) and its link function where the regression family is first described (Section 5.3), citing Agresti, A. (2002), and gloss "functional form."
> > - We also replaced the imprecise "no uniformly powerful CI test exists in high-dimensional settings" with the actual result of Shah & Peters (2020): “conditional independence is not testable without restrictions, which is why useful CI tests must constrain the null through modeling assumptions” (Section 5.3).
> >
> > $\textbf{Comment 15.}$ no reference for Bonferroni or FDR correction
> >
> > $\textbf{Response:}$ Good point, thanks.
> >
> > $\textbf{Action:}$ We now cite Dunn (1961) for the Bonferroni correction and Benjamini & Hochberg (1995) for FDR in Section 6.2.5.
> >
> > $\textbf{Comment 16.}$ Section 9.1 claims that gene expression data is continuous; but raw expression data is usually count data (nonnegative integers)
> >
> > $\textbf{Response:}$ Thank you for this helpful point.
> >
> > $\textbf{Action:}$ We now note that raw RNA-seq expression is count data (non-negative integers) and microarray measurements are continuous intensities, and that in practice counts are log- or variance-stabilising-transformed and then analysed as continuous (though often non-Gaussian and high-dimensional). This also reinforces the survey's mixed-type / discretization theme (Section 9.1).
> >
> > $\textbf{Comment 17.}$ Section 10.1 mentions sex as a binary variable, but that's not really the modern scientific understanding of it (for example, see this recent article, which also discusses the statistical issues that arise from such an assumption)
> >
> > $\textbf{Response:}$ Agreed.
> >
> > $\textbf{Action:}$ We are now including a more appropriate binary example (mutation (present, absent)) (Section 10.1).

---

> > > ### Author Response · Authors · 2026-06-09
> > > **Response to Reviewer fALU: Minor (Part 1)**
> > >
> > > (Continuing our point-by-point response.)
> > >
> > > $\textbf{Comment 1.}$ In the first paragraph in the intro, why are "correlations between molecules" in particular mentioned as biological research? Seems obvious that a lot of biological research is concerned with larger scale phenomena than molecular interactions.
> > >
> > > $\textbf{Response:}$ Thank you, we have broadened the phrasing accordingly.
> > >
> > > $\textbf{Action:}$ Broadened to "correlations among measured variables" in the introduction (Section 1).
> > >
> > > $\textbf{Comment 2.}$ Could be worth mentioning RESIT and other additive noise model approaches that also use CI tests.
> > >
> > > $\textbf{Response:}$ Thank you, this is a useful addition to the survey.
> > >
> > > $\textbf{Action:}$ Added a brief note that additive noise models such as RESIT also rely on independence tests, and they are applied to regression residuals. We have also added a citation to Peters et al. (2014) that introduced RESIT (Section 4).
> > >
> > > $\textbf{Comment 3.}$ Methods should be given citations when they're introduced, like PC, GS, and FCI at the end of Section 2.
> > >
> > > $\textbf{Response:}$ Thanks for the feedback.
> > >
> > > $\textbf{Action:}$ Added citations at first mention: PC (Spirtes & Glymour 1991), Grow–Shrink (Margaritis 2003), and FCI (Spirtes et al. 1995) (Section 2).
> > >
> > > $\textbf{Comment 4.}$ Many of the references are formatted incorrectly (capitalization like "Llm" and "markov"; "et al." instead of full lists of names; Pena instead of Peña, etc.)
> > >
> > > $\textbf{Response:}$ Thank you for the careful reading of the bibliography.
> > >
> > > $\textbf{Action:}$ We have carefully fixed all minor typo/formatting issues in the bibliography: brace-protected capitalization (e.g., LLM, Markov, acronyms), restored full author lists, and fixed accents (Peña, etc.).
> > >
> > > $\textbf{Comment 5.}$ Maybe swapping the sections 3.4 and 3.3 makes sense?
> > >
> > > $\textbf{Response:}$ We agree this ordering reads more naturally.
> > >
> > > $\textbf{Action:}$ The Markov Blanket section now precedes the Assumptions section, so Section 3 reads DAGs / Bayesian networks → CPDAGs → Markov blanket → assumptions, flowing more naturally into the algorithms.
> > >
> > > $/textbf{Comment 6.}$ In section 4, be more clear that it starts at a complete undirected graph
> > >
> > > $\textbf{Response:}$  Good point. The starting point should be stated explicitly.
> > >
> > > $\textbf{Action:}$ The opening now says the algorithms start from a fully connected undirected graph (Section 4).
> > >
> > > $\textbf{Comment 7.}$ I don't really get the "interpretability" argument repeated a few times about constraint-based methods. Doesn't this apply to GES too? Decomposability and local consistency of the score ensure each edge addition or removal has a clear interpretation, analogous to a CI test.
> > >
> > > $\textbf{Response:}$ Thank you for this thoughtful point.
> > >
> > > $\textbf{Action:}$ We have rewritten the relevant passage in Section 8 so that it no longer contrasts constraint-based methods with score-based.
> > >
> > > $\textbf{Comment 8.}$ PC was introduced in 1991 by this paper
> > >
> > > $\textbf{Response:}$ Thank you very much for the pointer and for providing this valuable reference.
> > >
> > > $\textbf{Action:}$ We now cite Spirtes & Glymour (1991) as the origin of PC, at its first mention.

---

> > > ### Author Response · Authors · 2026-06-09
> > > **Response to Reviewer fALU: Minor (Part 2)**
> > >
> > > (Continuing our point-by-point response.)
> > >
> > > $\textbf{Comment 9.}$ Section 5.5 includes the tautology "they can capture nonlinear relationships without assuming linearity"
> > >
> > > $\textbf{Response:}$ Thank you, we have reworded the sentence.
> > >
> > > $\textbf{Action:}$ Changed to "detect nonlinear conditional dependence without committing to a parametric functional form or Gaussianity.” (Section 5.5)
> > >
> > > $\textbf{Comment 10.}$ Inconsistent use of periods at the end of (sub(sub))section titles, e.g.., 6.2.5 vs 6.2.6
> > >
> > > $\textbf{Response:}$ Thank you for catching this.
> > >
> > > $\textbf{Action:}$ We standardized this; trailing periods removed from subsection and subsubsection titles.
> > >
> > > $\textbf{Comment 11.}$.In Section 8.2, what does "evaluate counterfactual scenarios, and assess external validity" mean?
> > >
> > > $\textbf{Response:}$ Thank you for the point raised.
> > >
> > > $\textbf{Action:}$ To improve the clarity of the statement we now state only what the structure supports: epistemic explanations are "required to anticipate the effects of interventions, which purely associational models cannot" (Section 8.2).
> > >
> > > $\textbf{Comment 12.}$ use \citet for the Agrahari ref in Section 9.2
> > >
> > > $\textbf{Response:}$ Thank you for this comment.
> > >
> > > $\textbf{Action:}$ We now use \citet for the Agrahari (2018)  reference (Section 9.2).
> > >
> > > $\textbf{Comment 13.}$ 3 different real data benchmarks are mentioned in Section 9.5, but only the last is claimed to "complement purely synthetic benchmarks"?
> > >
> > > $\textbf{Response:}$ Thank you for noting this; we have clarified it.
> > >
> > > $\textbf{Action:}$ We now state that all three real benchmarks complement the synthetic ones by grounding evaluation in real or controlled data with known structure, and we keep the distinctive feature of CausalChambers (variables set by direct physical intervention) (Section 9.5).
> > >
> > > $\textbf{Comment 14.}$ How about mentioning other powerful model-free tests, like distance correlation and Chatterjee's correlation?
> > >
> > > $\textbf{Response:}$ A worthwhile addition indeed.
> > >
> > > $\textbf{Action:}$ Added a note in the Section 5 that “powerful model-free measures of unconditional dependence (distance correlation (Székely et al. 2007) and Chatterjee's ξ (2021)) fall outside the survey's conditional-independence scope, though conditional extensions can in principle serve as CI tests.”
> > >
> > >
> > > We thank the reviewer for the careful and constructive feedback. We have addressed all the points raised, and we think that these revisions have made the manuscript clearer and more precise. We believe the revised manuscript is stronger as a result, and we are grateful for the time and attention the reviewer devoted to it.

---

### Review · Reviewer_B3n9 · 2026-05-29

**Summary Of Contributions:**

The paper reviews Conditional Independence (CI) tests for constraint-based causal discovery and provides a comprehensive survey of the assumptions required by different types of CI tests. It discusses their robustness, scalability, computational complexity, statistical power, advantages and limitations relative to alternative CI tests, as well as the software packages in which they can be practically implemented.

Building on this overview, the paper further investigates the consequences of Type I/II errors and qualitatively analyzes their impact on graph-level errors. It also presents practical considerations for CI testing, including discretization strategies and guidelines for selecting appropriate CI tests, making it a useful reference handbook.

In addition, the paper discusses the scalability and interpretability of CI tests and concludes with a discussion of their applications in biomedical domains.

**Audience:**

Yes

**Audience Explanation:**

The paper will be a valuable resource for researchers who are new to constraint-based causal discovery algorithms and CI tests. The discussions of graph-level errors and scalability are particularly informative. Although the paper does not introduce substantially new perspectives on these topics, it brings together several important yet often fragmented discussions in the literature and presents them in a comprehensive manner.

The key issue is the lack of references supporting these discussions, such as those concerning multiple-testing correction discussed above, as well as other empirical claims made throughout the paper.

**Claims And Evidence:**

Yes

**Claims Explanation:**

The paper is well written and organized within a clear framework. It provides a comprehensive review of CI tests for constraint-based causal discovery. The discussion covers statistical power, graph-level errors, scalability, and interpretability. The advantages and limitations of each CI test are presented clearly.

However, there are several aspects that could be improved:

1. In Section 6 and later sections, citations are limited when making statements such as "empirical studies suggest that..." or similar claims. Claims based on empirical observations should be supported by references. For example, in Section 6.2.5, which studies support the claim that the algorithms are self-adjusting without multiple-testing correction? Similarly, in Section 9.1, references should be provided for the statement that "causal and Bayesian network approaches have been applied extensively to transcriptomic datasets." It is recommended that the authors verify such claims throughout the manuscript and provide references to support them.

2. Some subsections in Section 6 are very short and seem to be placed inappropriately. For instance, Section 6.2.6 is more of a summary and does not contain sufficient new content to warrant a standalone subsection.

3. The organization of the survey could be improved further. Since the primary focus of the paper is CI testing, the introduction to each CI test could be presented in greater detail, including representative mathematical formulations where appropriate. In comparison, the discussion of CI tests is less extensive than the introduction to constraint-based causal discovery algorithms, which somewhat shifts attention away from the central topic of the survey.

Some minor issues:

1. In the causal discovery literature, the term "heterogeneous" often refers to data generated from different structural causal models rather than different data types. The paper sometimes uses "heterogeneous" to describe mixtures of different data types, which may lead to confusion. It would be helpful to standardize the terminology, especially if "mixed data" is the term more commonly used throughout the manuscript.

2. In Section 3.3.2 on Faithfulness, the assumptions of jointly independent noise terms and noise variables being independent of the observed variables should be stated explicitly when concluding that $X$ is marginally independent of $Z$.

3. Table 3 appears to contain an error. The table should describe FCI algorithms, but its content still refers to PC algorithms and duplicates Table 2.

**Requested Changes:**

1. Please review the manuscript carefully and ensure that all empirical observations and claims are supported by references.

2. It is recommended to provide a more detailed introduction to the CI tests, including representative mathematical formulations where appropriate.

3. Some subsections could be reorganized to avoid overly short subsections and improve the overall flow of the presentation.

---

> ### Author Response · Authors · 2026-06-09
> **Response to Reviewer B3n9 (Part 1)**
>
> We thank the reviewer for the careful reading and for the encouraging assessment: that the paper is "well written and organized within a clear framework," provides "a comprehensive review of CI tests," presents "the advantages and limitations of each CI test... clearly," makes "a useful reference handbook," and will be "a valuable resource for researchers who are new to constraint-based causal discovery algorithms and CI tests." We have addressed each point below; the reviewer's comments are quoted.
>
>
> $\textbf{Comment 1.}$ In Section 6 and later sections, citations are limited when making statements such as "empirical studies suggest that..." or similar claims. Claims based on empirical observations should be supported by references. For example, in Section 6.2.5, which studies support the claim that the algorithms are self-adjusting without multiple-testing correction? Similarly, in Section 9.1, references should be provided for the statement that "causal and Bayesian network approaches have been applied extensively to transcriptomic datasets." It is recommended that the authors verify such claims throughout the manuscript and provide references to support them.
>
> $\textbf{Response:}$ We thank the reviewer for pressing on this, which was well taken.
>
> $\textbf{Action:}$ On Section 6.2.5: the original "self-adjusting" statement reflected a remark in teaching material rather than a primary result, and on searching the literature we were unable to find any study establishing it; the literature in fact develops explicit error-rate control for PC precisely because uncorrected multiplicity is treated as a real concern. We therefore removed the unsupported statement and rewrote the paragraph around results we can cite: the significance level \alpha is a tuning parameter rather than a family-wise error rate, with high-dimensional consistency obtained by shrinking \alpha with the sample size (Kalisch & Bühlmann, 2007); controlling the per-test type I rate does not by itself control the graph-level error rate, so where explicit control is desired the Benjamini–Yekutieli  (Benjamini & Yekutieli, 2001), procedure is used, as in FDR-controlled PC (Strobl et al., 2019; Li & Wang, 2009) and false-discovery bounds for local learning (Tsamardinos & Brown, 2008). On Section 9.1, we added supporting references (Friedman, 2004; Markowetz & Spang, 2007). We also audited the manuscript for empirical claims and ensured the remaining ones carry citations.
>
> $\textbf{Comment 2.}$ Some subsections in Section 6 are very short and seem to be placed inappropriately. For instance, Section 6.2.6 is more of a summary and does not contain sufficient new content to warrant a standalone subsection.
>
> $\textbf{Response:}$ Agreed.
>
> $\textbf{Action:}$ We removed the standalone Section 6.2.6 ("Implications for causal discovery") and folded its content into a brief closing paragraph of Section 6.2, so it now reads as a synthesis of the power-and-error discussion rather than a separate subsection. The remaining subsections of Section 6.2 are of comparable, adequate length.
>
> $\textbf{Comment 3.}$ The organization of the survey could be improved further. Since the primary focus of the paper is CI testing, the introduction to each CI test could be presented in greater detail, including representative mathematical formulations where appropriate. In comparison, the discussion of CI tests is less extensive than the introduction to constraint-based causal discovery algorithms, which somewhat shifts attention away from the central topic of the survey.
>
> $\textbf{Response:}$ Agreed, and we have strengthened the CI-test discussion, which is the survey's central topic.
>
> $\textbf{Action:}$ Section 5 previously described the six families largely in prose; we added a representative test statistic to each:
> - Fisher's z-transform and the $\sqrt{n-|Z|-3}$ statistic (partial correlation) (Section 5.1);
> - the $G^2$ and Pearson $\chi^2$ statistics (contingency-table) (Section 5.2);
> - the GLM coefficient null and likelihood-ratio statistic (regression) (Section 5.3);
> - the k-NN conditional-mutual-information estimator (nearest-neighbor) (Section 5.4);
> - the KCI trace statistic (kernel) (Section 5.5);
> - and the GCM normalized residual-product statistic (machine-learning-based) (Section 5.6).
>
> Each formulation is drawn from the corresponding primary source which is appropriately cited in our manuscript.

---

> > ### Author Response · Authors · 2026-06-09
> > **Response to Reviewer B3n9 (Minor Issues - part 2)**
> >
> > (Continuing our point-by-point response.)
> >
> > $\textbf{Comment 1.}$ In the causal discovery literature, the term "heterogeneous" often refers to data generated from different structural causal models rather than different data types. The paper sometimes uses "heterogeneous" to describe mixtures of different data types, which may lead to confusion. It would be helpful to standardize the terminology, especially if "mixed data" is the term more commonly used throughout the manuscript.
> >
> > $\textbf{Response:}$ Thank you for this helpful clarification.
> >
> > $\textbf{Action:}$ We checked every occurrence of "heterogeneous" and confirmed that each referred to a mixture of data types rather than to multiple data-generating models; we have replaced all such uses with "mixed" / "mixed-type" (Abstract, Section 2,  Section 5.5, and Section 9), consistent with the term used elsewhere in the paper.
> >
> > $\textbf{Comment 2.}$ In Section 3.3.2 on Faithfulness, the assumptions of jointly independent noise terms and noise variables being independent of the observed variables should be stated explicitly when concluding that $X$ is marginally independent of $Y$
> >
> > $\textbf{Response:}$ Agreed.
> >
> > $\textbf{Action:}$ In the path-cancellation example we now state explicitly that the noise terms  $\epsilon_Y$ and  $\epsilon_Z$ are jointly independent and independent of the observed variables. With these assumptions stated, the conclusion is clear: after the cancellation, $Z = \beta \epsilon_Y + \epsilon_Z$ is a function of the noise alone and is therefore marginally independent of X. Following reviewer's (Reviewer fALU) suggestion we reordered Sections 3.3 and 3.4 so that the Markov Blanket section now precedes Assumptions; the Faithfulness example is therefore now in Section 3.4.2 rather than in Section 3.3.2.
> >
> > $\textbf{Comment 3.}$ Table 3 appears to contain an error. The table should describe FCI algorithms, but its content still refers to PC algorithms and duplicates Table 2.
> >
> > $\textbf{Response:}$ Thank you very much for catching this.
> >
> > $\textbf{Action:}$ The FCI feature table had erroneously carried rows labeled with PC-specific names; we have renamed them to the corresponding FCI implementation features, so the table now describes FCI and no longer duplicates Table 2.
> >
> >
> > We thank the reviewer for the careful and constructive feedback. We have addressed all the points raised, and we think that these revisions have made the manuscript clearer and more precise. We believe the revised manuscript is stronger as a result, and we are grateful for the time and attention the reviewer devoted to it.

---

### Author Response · Authors · 2026-06-09
**Summary of revision**

We sincerely thank the Action Editor and the three reviewers for their careful and constructive evaluation.

We are grateful that all three evaluators found the survey useful and well-organized (e.g. "a helpful review paper that collects a lot of information … in one coherent place" (Reviewer fALU); a "comprehensive review of CI tests" that makes "a useful reference handbook" (Reviewer B3n9); and a resource of interest to "PhD students, as well as practitioners who need to identify the appropriate CI testing strategies" (Reviewer SyJS)).

The requested changes concerned missing definitions and citations, a number of correctness and precision points, and suggestions on depth and organization, all of which we have addressed. The main improvements we have made to the manuscript by following the reviewers’ feedback are the following:

$\textbf{Improved justification on supported claims}$ (Reviewers B3n9 and fALU). We revised statements that were not adequately supported, replacing them with claims backed by the literature and adding citations where empirical observations were made. We audited such claims throughout the manuscript and elaborate on the specific changes in our point-by-point reply (Section 6.2.5 and Section 9.1).

$\textbf{Enhanced Depth on CI tests}$ (Reviewer B3n9). We added a representative test statistic to each of the six CI-test families in Section 5 (partial-correlation (Section 5.1), contingency-table (Section 5.2), regression (Section 5.3), nearest-neighbor (Section 5.4), kernel (Section 5.5), and machine-learning-based (Section 5.6)), strengthening the conditional independence tests component of the survey.

$\textbf{Improved Organization of Paper}$ (Reviewers fALU and B3n9). We reordered Section 3 so the Markov-blanket material now precedes the assumptions (a more natural lead order) and merged  the short Section 6.2.6 into the surrounding discussion; specifics are in our point-by-point reply.

$\textbf{Added Necessary Definitions and References}$ (Reviewer fALU). We now define d-separation (Section 3.1), the skeleton (Section 3.2), the PAG (Section 3.4.3), GLM (Section 5.3), and Meek's rules (Section 4.2), citing a primary source where applicable (the skeleton, for instance, is defined inline without a citation), and added citations for CPDAGs (Section 3.2), stable-FCI (Section 4.3), the Markov-blanket algorithms (Section 4.4), Bonferroni (Section 6.2.5), and FDR (Section 6.2.5).

$\textbf{Correctness and precision}$ (Reviewers fALU and B3n9). We corrected the Faithfulness/Markov-condition wording (Sections 3.3, 3.3.2), made the noise assumptions explicit in the Faithfulness example (Section 3.3.2), fixed the Grow-Shrink CI-test statement and complexity (Section 4.4), corrected the gene-expression-as-continuous statement (Section 9.1), and corrected Table 3, which had erroneously listed PC rows in the FCI feature table (flagged by both Reviewers fALU and B3n9).



$\textbf{Improved Consistency}$ (Reviewers fALU and B3n9). We standardized terminology throughout, replacing "heterogeneous" with “mixed(-type)” wherever it referred to data types, and “collider” with “v-structure”.


We reply to each reviewer point-by-point in separate comments. All changes are highlighted in $\textcolor{blue}{blue}$ in the attached PDF.

---

### Author Response · Authors · 2026-06-09
**New revision available**

Dear action editor, dear reviewers,

We would like to express our gratitude for your careful and constructive evaluation.  We feel that the manuscript is much improved after following your suggestions.

We have uploaded the revised manuscript. The main changes are in Section 3.3.2 (Faithfulness assumptions), Section 5 (representative formulas for each CI-test family), Section 6.2.5 (multiple-testing discussion) and the merged Section 6.2.6, Section 9.1 (gene-expression data and added citations), and Table 3, together with the definition/citation additions and terminology standardization described in our point-by-point replies. All changes are highlighted in $\textcolor{blue}{blue}$.

---

### Decision · Action_Editor_DfKJ · 2026-07-03

**Recommendation:** Accept as is

**Audience:**

Yes

**Audience Explanation:**

This is a nice survey about the topic of constraint-based causal discovery, very useful to the community.

**Claims And Evidence:**

Yes

**Claims Explanation:**

This is a thoughtful survey with adequate citations.